# Truly Assessing Fluid Intelligence of Large Language Models through Dynamic Reasoning Evaluation

## Abstract

Recent advances in large language models (LLMs) have demonstrated impressive reasoning capacities that mirror human-like thinking. However, whether LLMs possess genuine fluid intelligence (i.e., the ability to reason abstractly and generalize rules in novel situations) remains an open question. Existing reasoning benchmarks either focus on domain-specific knowledge (crystallized intelligence) or lack interpretability. To address these limitations, we propose DRE-Bench, a dynamic reasoning evaluation benchmark grounded in a hierarchical cognitive framework. DRE-Bench consists of 36 abstract reasoning tasks organized across four cognitive levels, with each task featuring multiple dynamic variants that test the same underlying latent rule. This design enables fine-grained, interpretable, and reliable assessments of fluid intelligence. We evaluate a range of state-of-the-art LLMs, including both general LLMs (GPT-4o, Claude 3.7) and reasoning LLMs (o1, DeepSeek-R1, QwQ, Skywork-OR1). Experimental results reveal that although most LLMs achieve competent and robust performance in low-level cognition, they struggle with high-level cognition and exhibit limited generalization as task complexity grows. Our findings highlight the gap between current LLMs and true human-like fluid intelligence and offer a new path for systematically tracking reasoning progress in LLMs.

## 1 Introduction

Recently, large language models (LLMs) (OpenAI, 2024b; DeepSeek-AI et al., 2025; Anthropic, 2024; OpenAI, 2024a; Yang et al., 2024a) have achieved remarkable success across various applications, such as disciplines (Cobbe et al., 2021; Lewkowycz et al., 2022), intelligent chatbots (Zhang et al., 2023; Ouyang et al., 2022) and code generation (Chen et al., 2021; Nijkamp et al., 2023). Models like OpenAI's o1 (OpenAI, 2024b) leverage substantial test-time computation to refine their reasoning processes, learn from previous errors, and explore diverse strategies, exhibiting a degree of cognitive behavior that closely mirrors human-like thinking. As such, there is an urgent need for a principled evaluation framework to track and quantify the reasoning intelligence of cutting-edge LLMs systematically.

Existing reasoning benchmarks can be broadly categorized into two major types: crystallized intelligence (Cattell, 1963; Schipolowski et al., 2014) and fluid intelligence (Cattell, 1963; Kent, 2017). Crystallized intelligence refers to models' ability to apply accumulated knowledge to solve problems. Representative benchmarks such as AIME (Ye et al., 2025), GPQA (Rein et al., 2024), and SuperGPQA (Du et al., 2025) which require multi-step reasoning grounded in domain-specific knowledge. However, as LLMs increasingly achieve expert-level performance on such knowledge-intensive tasks, the community gradually recognized that fluid intelligence—the ability to generalize beyond memorized content and reason in novel settings—is becoming increasingly important (Raven, 2003; Flanagan et al., 2007). In assessing the fluid intelligence of LLMs, ARC-AGI series (Chollet, 2019; Chollet et al., 2024) raise abstract reasoning tasks and is regarded as a milestone. Such tasks require LLMs to infer the latent rule solely from provided input-output training pairs and generalize it to predict correct outputs for novel testing inputs. Figure 1(a) illustrates two examples of such latent rules, frequency identification and category classification.

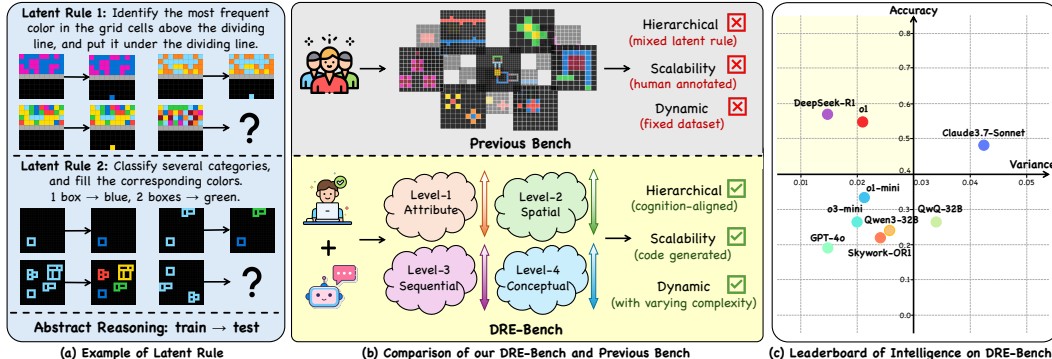

Figure 1: (a) Examples of the latent rule hidden in test cases. (b) Compared with previous benchmarks, our DRE-Bench demonstrates advantages in terms of hierarchy (cognition-aligned), scalability (code-generated), and dynamism (varying complexity). (c) Leaderboard of LLM intelligence on DRE-Bench, with accuracy on the y-axis and stability on the x-axis.

Although recent efforts (Chollet, 2019; Chollet et al., 2025) have attempted to measure the fluid intelligence of LLMs, such as analyzing atomic operations (Wu et al., 2025) and the stochastic parrot phenomenon (Yu et al., 2025), they face several limitations as shown in Figure 1(b). First, existing benchmarks usually comprise abstract reasoning cases whose latent rules are not linked with stages of human cognition (Primi, 2001). Consequently, it is hard to tell what level of human-like intelligence a model has reached. Second, previous studies require manual annotation, which is labor-intensive and constrains benchmarks' scalability and diversity of latent rules. Third, these benchmarks are inherently static, with each latent rule linked to only one or a few fixed cases. Such a static nature suffers from data contamination (Li et al., 2024a; Yang et al., 2024b), making it hard to determine whether the model truly understands the latent rule or merely memorizes it.

To address these challenges, we propose a Dynamic Reasoning Evaluation benchmark, DRE-Bench, designed to assess the genuine fluid intelligence of large language models (LLMs). DRE-Bench is structured around a confirmed psychology hierarchy (Primi, 2001), with four cognitive levels ranging from simple to complex reasoning: Attribute, Spatial, Sequential, and Conceptual level. Each level contains 3 latent rules specified by several designed abstract reasoning tasks. Due to the simple data format of abstract reasoning tasks, we design a code-based generator and solver for each task, which can generate multiple dynamic variants with different levels of complexity. In total, DRE-Bench provides about 4K abstract reasoning cases. This framework enables a fine-grained, cognition-aligned evaluation of the abstract reasoning ability and allows for a robust assessment of fluid intelligence by analyzing both accuracy and variance across tasks with consistent latent rules.

Compared to existing benchmarks, DRE-Bench offers three key advantages as illustrated in Figure 1(b). *i) Cognition-aware task hierarchy.* DRE-Bench presents reasoning tasks with a cognitive hierarchy, which explicitly aligns each task with four human-like cognitive levels. This alignment provides good interpretability and allows mapping model behavior to specific cognitive capabilities. *ii) Human-Agent Collaboration Pipeline.* For each latent rule, we employ LLM-driven agents to design code-based generators and solvers, which can produce input samples and corresponding answers accurately. To this end, our data generation pipeline achieves high correctness, efficiency, and scalability. *iii) Dynamic evaluation.* DRE-Bench supports dynamic generation of multiple task instances by flexibly varying the latent rule-related variables, obtaining extensive variants with different levels of complexity. This dynamic property helps avoid the data contamination issue that static datasets are prone to (Li et al., 2024a;b; Yang et al., 2024b). Therefore, we can precisely and comprehensively assess whether LLMs have truly grasped the underlying reasoning rules, further tracking the fluid intelligence of current LLMs.

We conduct comprehensive experiments on DRE-Bench using a range of LLMs, including general-purpose models without explicit reasoning capabilities such as GPT-4o (OpenAI, 2024a) and Claude-3.7 (Anthropic, 2024), and reasoning LLMs (models with thinking) such as OpenAI-o1 (OpenAI, 2024b), DeepSeek-R1 (DeepSeek-AI et al., 2025), QwQ (Yang et al., 2024a), Skywork-OR1 (He et al., 2025), etc. The takeaways of our key findings are as follows:

- As the cognitive level of the reasoning tasks increases, model accuracy consistently declines, particularly for tasks involving physical concepts. Among them, OpenAI-o1 and DeepSeek-R1 demonstrate stronger performance and stability, while Claude 3.7 stands out in general LLMs. (Figure 1(c) and Section 4.2).

- Reasoning LLMs outperform general LLMs on most abstract reasoning tasks. Moreover, as the cognitive level increases, the difference between models becomes more pronounced: differences may be minimal on lower-level tasks, but in higher-level tasks, stronger LLMs will exhibit a more obvious advantage (Section 4.2).

- We analyzed model accuracy and stability across different complexities. We observed that with the complexity of a specific task increasing, models whose performance declines may not possess genuine fluid intelligence; only those that continue to perform well can be considered to truly master the underlying reasoning rules (Section 4.3).

- Increasing the number of in-context training examples can slightly boost LLMs' performance. However, adding visual information about the abstract reasoning problems has little positive impact, and sometimes even leads to a decrease in model accuracy (Section 4.4).

- Inference time scaling plays a more important role in low-level reasoning tasks, but may be insufficient towards high-level latent rules as complexity increases (Section 4.4).

Overall, the **contributions** of this paper are summarized as follows. 1) We propose an abstract reasoning benchmark with a cognition hierarchy, providing a more structural and comprehensive system to analyze the LLMs' true fluid intelligence. 2) We develop a verifiable and scalable data engine to dynamically generate abstract reasoning data with various complexities, by designing a generator and solver for each task. 3) We perform comprehensive evaluations on a variety of popular LLMs, indicating that the existing LLMs still struggle to solve the reasoning problem of high cognitive levels. Existing LLMs may not have truly internalized the underlying reasoning rules, which highlights that they remain far from achieving true fluid intelligence.

## 2 RELATED WORK

### 2.1 EVALUATION FOR FLUID INTELLIGENCE

There have been numerous attempts to define and measure the intelligence degree of existing large language models. Among them, the Abstraction and Reasoning Corpus(ARC) (Chollet, 2019) is regarded as a milestone, which defines that true intelligence should possess skill-acquisition efficiency. This concept attracted broad attention and led to many analytical studies (Wu et al., 2025; Yu et al., 2025; Acquaviva et al., 2022; Xu et al., 2023; Wang et al., 2023; 2024a). (Wu et al., 2025) select some atomic abstract reasoning operations, and find that LLMs perform poorly on some atomic operations. (Yu et al., 2025) designed PHYSICO to evaluate whether LLMs really understand the physical phenomena they describe, by comparing language-format description and corresponding ARC format grid. However, existing abstraction reasoning benchmarks haven't categorized tasks along cognitive dimensions, and can only provide a coarse-grained evaluation of LLMs' reasoning ability. In addition, all these benchmarks are static, implying that they are highly susceptible to data contamination and only possess fixed complexity. Therefore, our work proposes DRE-Bench, a hierarchical cognitive dynamic benchmark on abstract reasoning. DRE-Bench can automatically generate data with varying levels of complexity, enabling comprehensive and fine-grained evaluation of LLM intelligence.

### 2.2 DYNAMIC EVALUATION

Studies (Li et al., 2024a;b; Yang et al., 2024b) have found that static benchmarks are highly prone to data contamination and have detected severe data contamination rates in some LLM benchmarks like (Wang et al., 2018; 2024b). Moreover, their static nature implies a fixed level of complexity, making it difficult to adapt to evolving model capabilities. Therefore, some researchers have pioneered the exploration of dynamic evaluation on LLMs. Study (Zhu et al., 2023) proposed DyVal to dynamically generate test samples based on the graph structure to combat data contamination. Similarly, NPHardEval (Fan et al., 2023) generates new evaluation samples for NP-hard mathematical problems. To extend dynamic evaluation to more diverse NLP tasks, (Zhu et al., 2024) further

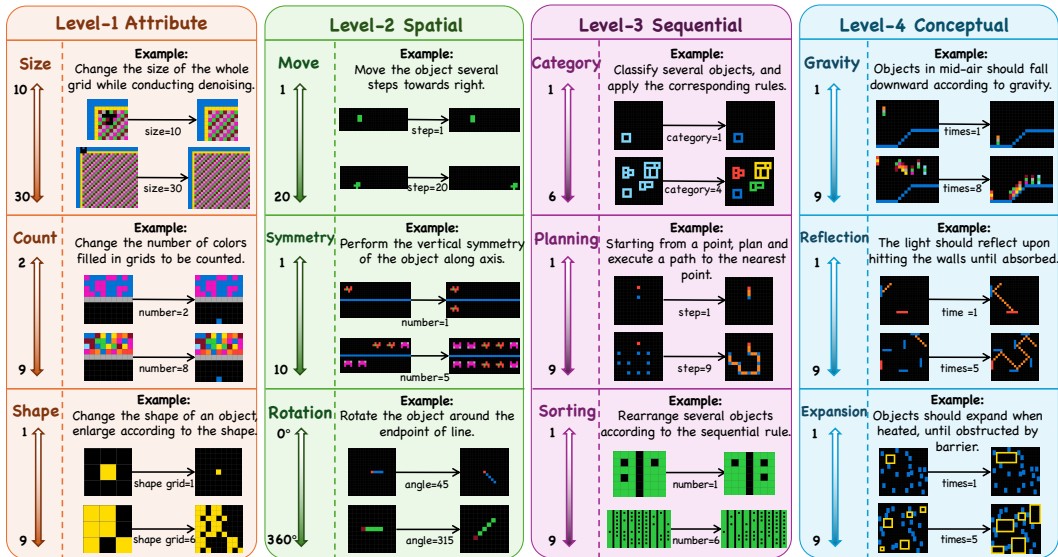

Figure 2: Specific abstract reasoning tasks across four cognitive levels. For each task, we visualize two pairs of input and output, corresponding to two different values of the dynamic variable. The arrows are labeled with variable ranges, with darker colors indicating higher complexity.

developed MPA, which employs LLM-based agents to transform existing problems into new ones. However, most of these dynamic evaluation methods are designed for general NLP tasks and are not applicable to more complex reasoning scenarios. More critically, the accuracy of their dynamically generated data is difficult to verify, leaving their reliability in constant doubt. In this work, we are the first to introduce a dynamic evaluation paradigm for abstract reasoning tasks. Our data generation process is code-verifiable, ensuring 100% reliability of the generated samples.

## 3 METHOD

### 3.1 CONSTRUCTING COGNITION-INSPIRED ABSTRACT REASONING FRAMEWORK

Studies about fluid intelligence (Raven, 2003; Carpenter et al., 2018; Primi, 2001) indicate that the complexity of a reasoning problem may be related to the types of rules applied in the inductive reasoning process. Among them, the rule-type hierarchy proposed by Ricardo (Primi, 2001) represents a relatively comprehensive cognitive framework in psychology. This framework categorizes inductive rule-type as four top-down levels, and proves the four levels form a true cognitive hierarchy: as from rule level 1 to 4, people impose qualitatively greater demands on abstraction, working memory, with reaction times and error rates also increasing. Therefore this categorization is suitable to assess the human-like fluid intelligence of LLMs.

According to this cognitive hierarchy of reasoning rule and corresponding rule variables, we propose our abstract reasoning framework as Figure 2. For the first-tier framework, we adopt four levels, namely (1) Attribute, (2) Spatial, (3) Sequential, and (4) Conceptual. Then, for each cognitive level, we summarize a series of related rule variables related to abstract reasoning tasks. Finally, for each rule variable, we design three sets of dynamic case generators to enable fine-grained evaluation of LLMs' corresponding cognitive reasoning capabilities. The detailed dataset table is in Appendix C.

**Level-1: Attribute.** In the attribute level, we follow the operational dimensions identified in cognitive psychology (Primi, 2001), dynamically evaluating the reasoning capabilities of LLMs along three key rule types: size, count, and shape.

**Level-2: Spatial.** In the spatial level, drawing on psychological studies, we designed a set of classic rules that comprehensively capture the notion of spatial reasoning, namely move, rotation, and symmetry. Specifically, for the "move" rule, we design dynamic data along five directional axes: up, down, left, right, and upper-right. For each direction, we set the moving distance from 1 to 30. This enables a fine-grained assessment of the LLM's understanding of both moving direction and

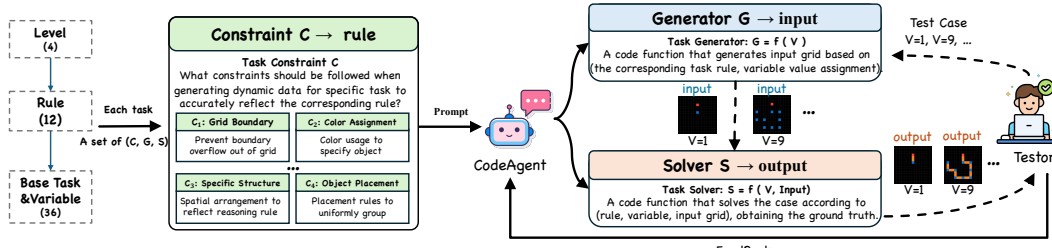

Figure 3: DRE-Bench Data Generation Pipeline: (1) Professionals identify task-specific constraints and rules. (2) A CodeAgent collaborates with annotators to implement the generator and solver. (3) Different configurations are used to produce diverse cases.

distance. Similarly, for the "rotation" rule, we design two types of rotation axes, namely around an endpoint and around the center of objects. For each rotation setting, we change the rotation angle from 0 to 360 degrees. For the "symmetry" rule, we design tasks based on horizontal, vertical, and diagonal symmetry. For each type, the number of objects to be symmetrized can vary arbitrarily.

**Level-3: Sequential.** For Level-3, we incorporate reasoning rules that require multi-step inference and higher-order abstract ability. Specifically, we include: category learning, which requires identifying categories based on shared attributes across varying contexts; sorting, which requires understanding order and rearranging placement; and planning, which involves goal-directed problem solving by multiple reasoning steps. To precisely control task complexity within these reasoning types, we designed corresponding rule variables: the number of categories to be distinguished, the number of elements to be sorted, and the number of planning steps required.

**Level-4: Conceptual.** For Level-4, we focus on scientific concepts, which require not only high-level abstract reasoning but also the application of conceptual knowledge. Drawing inspiration from fundamental branches of physics (Yu et al., 2025), we introduce three representative concepts: gravity, reflection, and expansion. To further increase task complexity, we progressively intensify the application of these physical rules.

## 3.2 DATA GENERATION FRAMEWORK

After determining the cognitive level, we proceed to select the specific rule to evaluate the LLM's reasoning performance. To enable fine-grained assessment, we design approximately three tasks for each rule. For example, the "move" rule includes five directional tasks: up, down, left, right, and upper-right movement. As shown in Figure 3, for each task, we identify its underlying constraint, then a code agent constructs a set of generators and solvers, upon human inspection, can be used to batch-produce input-output pairs. Such a human–agent collaboration pipeline can ensure scalability not only in the volume of data but also in the diversity of new rule.

**Identifying Constraint.** First, for a given task, professionals identify all case-relevant constraints, such as <grid boundary>, <color assignment>, <object placement>, and so on. These constraints, together with the corresponding rule, are then transformed into structured prompts, where a dynamic variable is explicitly defined. Each prompt subsequently invokes a code agent to generate two functions(i.e., a generator and a solver) parameterized by the dynamic variable.

**Producing Generator and Solver.** In the second step, an LLM-driven code agent is employed to implement the generator and solver functions for each task. Based on the rule and constraints encapsulated in the prompt (example in Appendix D), the code agent produces a generator that serves to generate the input grid, with a tunable parameter controlling the complexity of input cases. The paired solver is also implemented to parse the input grid and generate the corresponding ground-truth output grid. To ensure the correctness of the generator and solver, we predefine a set of parameter configurations to verify consistency between the input and output grids. If the generator–solver pair passes manual inspection, it is retained; otherwise, the code agent is re-invoked for refinement until a valid pair is produced. A random seed is embedded in the generator to enable scalable and reproducible generation of an unbounded number of diverse, constraint-satisfying samples.

**Data Generation.** Once the final generator and solver are established, for each rule, we can configure various parameters and different random seeds to generate batches of cases with varying levels

Table 1: Model performance across four cognitive reasoning levels and corresponding tasks.

| | Level 1 Attribute | | | | Level 2 Spatial | | | | Level 3 Sequential | | | | Level 4 Conceptual | | | |
|---|---|---|---|---|---|---|---|---|---|---|---|---|---|---|---|---|
| Model | Size | Count | Shape | Avg-1 | Rotation | Move | Symmetry | Avg-2 | Category | Sort | Planning | Avg-3 | Optics | Mechanics | Thermal | Avg-4 |
| *General LLMs* | | | | | | | | | | | | | | | | |
| Claude-3.7 | 65.22 | 63.14 | 13.33 | 58.76 | 68.57 | 57.80 | **49.33** | 58.43 | **54.44** | 2.50 | **54.44** | 44.05 | **8.00** | 15.87 | 0.00 | 7.96 |
| Qwen3-32B | 61.79 | **71.05** | 18.33 | 60.05 | 51.43 | 29.20 | 1.33 | 27.66 | 7.69 | 3.75 | 8.89 | 7.14 | 0.00 | 0.00 | 0.00 | 0.00 |
| GPT-4o | 62.81 | 44.48 | 13.33 | 51.2 | 27.30 | 3.80 | 2.67 | 9.9 | 8.89 | 2.50 | 8.89 | 7.61 | 0.00 | 0.00 | 0.00 | 0.00 |
| Qwen2.5-32B | 44.72 | 28.42 | 6.67 | 35.06 | 5.71 | 0.20 | 0.00 | 1.65 | 4.62 | 1.25 | 7.78 | 4.57 | 0.00 | 0.00 | 0.00 | 0.00 |
| *Reasoning LLMs* | | | | | | | | | | | | | | | | |
| o1 | 64.75 | 60.00 | 58.33 | 62.45 | **93.08** | 69.69 | 6.67 | 58.88 | 26.67 | **11.25** | 53.33 | 28.92 | 0.00 | 7.94 | 0.00 | 2.65 |
| DeepSeek-R1 | 60.83 | 69.43 | 8.33 | 57.86 | 82.72 | **78.90** | 16.00 | **62.79** | 44.44 | 0.00 | 44.44 | 35.55 | 0.00 | 1.59 | 0.00 | 0.53 |
| o1-mini | 40.33 | 65.43 | 18.33 | 46.25 | 63.04 | 32.10 | 0.00 | 31.78 | 43.33 | 7.50 | 43.33 | 36.16 | 0.00 | 0.00 | 0.00 | 0.00 |
| o3-mini | 31.48 | 60.10 | **71.67** | 45.49 | 50.14 | 20.00 | 1.33 | 23.13 | 25.56 | 7.50 | 25.56 | 21.95 | 0.00 | **31.75** | 0.00 | **10.58** |
| QwQ-32B | **78.59** | 61.05 | 13.33 | **65.49** | 64.76 | 22.80 | 4.00 | 29.12 | 12.31 | 0.00 | 34.44 | 14.27 | 0.00 | 0.00 | 0.00 | 0.00 |
| SkyWork-OR1-32B | 59.62 | 68.95 | 13.33 | 57.59 | 64.76 | 15.90 | 4.00 | 25.98 | 9.23 | 0.00 | 36.67 | 12.87 | 0.00 | 0.00 | 0.00 | 0.00 |
| *Average vs Human* | | | | | | | | | | | | | | | | |
| Model-avg | 57.01 | 59.21 | 23.50 | 46.57 | 57.15 | 33.04 | 8.53 | 32.91 | 23.72 | 3.63 | 31.78 | 19.71 | 0.80 | 5.72 | 0.00 | 2.17 |
| Human-avg | 75.96 | 82.02 | 71.72 | 77.51 | 84.65 | 77.78 | 44.25 | 70.38 | 75.75 | 29.49 | 89.90 | 65.05 | 49.68 | 76.16 | 16.16 | 47.33 |

of complexity. This data generation pipeline not only extends to large amounts of data with high correctness, but also ensures scalability to conveniently integrate new rules.

## 4 EXPERIMENTS

In this section, we evaluate state-of-the-art large language models and investigate the following research questions through experimental results: i) How do current LLMs perform in abstract reasoning across different cognitive levels? (Section 4.2); ii) As the complexity of dynamic data increases, how will the LLM's performance change? (Section 4.3); iii) Based on the performance of different LLMs across various cognitive dimensions, to what extent has the model's intelligence level reached? (Section 4.3); iv) Is inference time scaling, visual information, and number of training context samples, truly effective for abstract reasoning tasks? (Section 4.4).

### 4.1 EXPERIMENTAL SETTINGS

**Evaluated LLMs.** For completeness, we test 11 representative LLMs varying in parameters, vision encoders, including close-sourced APIs and open-sourced LLMs. Close-sourced APIs from different companies encompass GPT-4o (OpenAI, 2024a), OpenAI-o1 (OpenAI, 2024b), Claude-3.7 (Anthropic, 2024) and OpenAI-o3-mini (OpenAI, 2025). Open-sourced LLMs include DeepSeek-R1 (DeepSeek-AI et al., 2025), QwQ, Qwen2.5 (Yang et al., 2024a), and Skywork-OR1 (He et al., 2025). See Supplementary Materials for details of evaluated LLMs. To reduce randomness, all presented results of models are average results over three trials.

**Evaluation Methods.** In the DRE-Bench benchmark, accuracy serves as the primary evaluation metric, defined as the proportion of samples for which the model's output grid exactly matches the ground-truth output grid. To avoid contingency, each variable contains 12 samples for each value on average. All inferences are performed using the vLLM backend (Kwon et al., 2023). To ensure fairness and consistency, we adopt the official standardized prompting template released by ARCPrize (Prize, 2024).

### 4.2 MAIN RESULTS IN FOUR LEVELS

Based on the defined cognitive levels from psychology, we first evaluate model performance at each level. The main results are presented in Table 1. Overall, as the cognitive level increases, model performance exhibits a clear downward trend, which aligns with established rules in human cognitive development. Among general LLMs, Claude-3.7 consistently achieves the highest performance across all levels. Notably, it performs well even on Level 3 tasks, where many models struggle significantly. When comparing general-purpose models with reasoning-specialized models, the latter consistently outperform the former in terms of average cognitive level. Among the reasoning models, both OpenAI-o1 and DeepSeek-R1 demonstrate clear advantages. A substantial performance gap is observed between vanilla LLMs and reasoning-enhanced LLMs—for example, QwQ-32B versus Qwen2.5-32B—showing an average difference of over 20%.

Furthermore, as task difficulty increases, performance disparities among models become more pronounced, highlighting the potential of incorporating dedicated reasoning paradigms for addressing fluid intelligence problems. For Level 4 tasks, which require conceptual knowledge, all existing

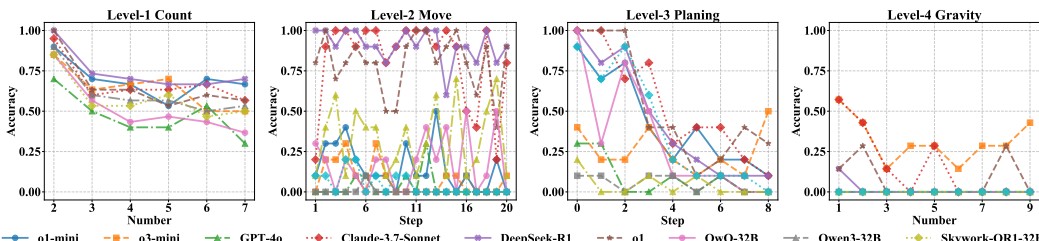

Figure 4: Model performance curves under varying complexities in four cognitive reasoning levels.

models fail, underscoring the current limitations of even advanced reasoning models. These findings emphasize both the inherent challenges posed by our benchmark and its flexibility in revealing model capabilities across a wide spectrum of cognitive demands.

What's more, we conduct a human study to validate our cognitive-aligned data framework. We extract 10% samples(about 400) from DRE-Bench based on its data distribution, and release a questionnaire to 40 professional annotators covering 19-50 age ranges. They are requested to fill out the test output as LLMs evaluated. Specifically, we provided a salary of 30 dollars per hour to each participant. The detailed age distribution of participants, UI interface, and readme instructions are in Appendix E.4. We can observe in Table 1 that human accuracy also generally decreases as the level increases, which validates the justification of our 4-level framework. Compared with LLMs, human accuracy is slightly higher on average, indicating that existing LLMs have not yet reached human-level abstract reasoning, which is consistent with studies (Chollet, 2019; Chollet et al., 2025). Furthermore, we conducted an independent t-test on the distributions of models and humans in Appendix Table 9. It demonstrates the statistical significance of humans' and models' results on DRE-Bench, further validating the data framework.

### 4.3 DYNAMIC TRENDS ACROSS DIFFERENT COGNITIVE LEVELS

Since our generator is capable of producing data with varying levels of complexity, we conduct a fine-grained evaluation to assess model performance across data with different complexity. Figure 4 illustrates representative performance curves of nine LLMs for each cognitive level, with cases under the same rule gradually increasing in difficulty. More task curves are provided in Appendix E.3.

As Figure 4, since tasks on the `Level-1 Attribute` involve basic enumeration without substantial cognitive demands, most models consistently achieved high average accuracy, and increases in complexity had minimal impact. As for `Level-2 Spatial`, performance differences among models became increasingly pronounced, lower-performing models continued to struggle with even simple cases. Impressively, models with high accuracy remained robust, relatively unaffected by the increase in case complexity. This suggests that these models have, to some extent, acquired the capability to resolve spatial reasoning problems. Regarding tasks in `Level-3 Sequential`, we observe a substantial performance drop as the number of required planning steps increases. Most models can only manage the simplest scenarios, with a consistent failure point emerging when the planning depth reaches two steps. This highlights that current LLMs remain limited in intelligence and have yet to truly master such sequential rules. Finally, at `Level-4 Conceptual`, almost all models fail to provide correct solutions, even in the simplest cases under the gravity rule, indicating that current models have only a rudimentary grasp of physical concepts and have yet to internalize even the most fundamental principles of intuitive physics. In general, as task complexity increases across each cognitive level, the accuracy of models tends to decrease or fluctuate accordingly.

To further illustrate the performance and stability of each model on dynamic task variants, Figure 5 presents the mean accuracy and corresponding variance across different cognitive levels. Here, we choose several top-performing models to analyze: o1, Claude 3.7, DeepSeek-R1, and QwQ-32B. As shown in the figure, for the majority of `Level-1` attribute tasks, OpenAI-o1, DeepSeek-R1, and Claude-3.7 demonstrate strong performance and high stability. However, when the task level increases to `Level-2` spatial, Claude-3.7 exhibits substantial fluctuations in performance, indicating limited generalization capabilities at this level. In contrast, OpenAI-o1 and DeepSeek-R1 maintain comparable performance and stability to those observed at `Level-1`, highlighting the advantage of reasoning models in solving more cognitively demanding tasks. Moreover, in `Level-3` sequential, most of the scatter points are concentrated in the lower-left region, suggesting

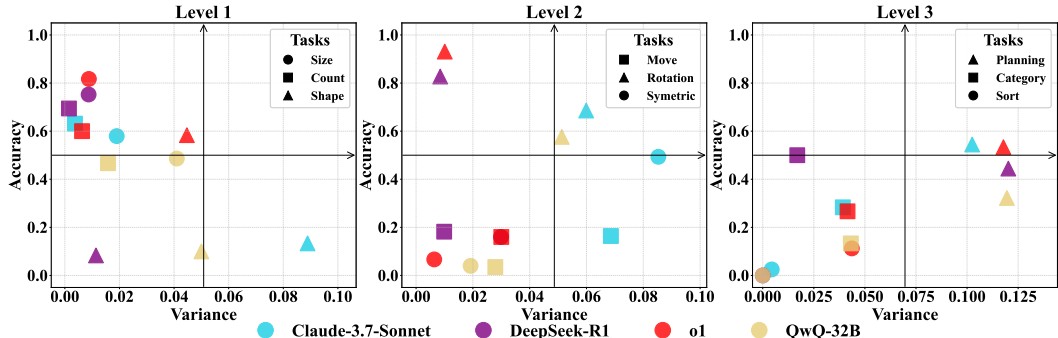

Figure 5: Scatter plots of model accuracy versus variance in cognitive reasoning levels and corresponding tasks, where points closer to the upper-left indicate higher accuracy and greater stability.

that current models struggle to generalize effectively across the more complex and varied tasks at higher levels. The full results of accuracy versus variance on all evaluated models in Figure 14.

## 4.4 ABLATION STUDY

**Impact of the Number of In-context Learning Samples.** Previous work (Brown et al., 2020; OpenAI, 2023; Work) has demonstrated the effectiveness of in-context learning in enhancing models' performance across some LLM tasks. Therefore in this section, we investigate how the quantity of in-context samples affects performance in the abstract reasoning scenario. The average results of all evaluated models are shown in Figure 6.

Overall, increasing the number of in-context samples helps models better capture underlying rules and improve performance. In higher levels like `Level-2 Spatial`, `Level-3 Sequential` and `Level-4 Conceptual`, increasing the number of in-context training samples leads to noticeable performance improvements. However, for `Level-1` tasks, increasing the number of samples yields limited improvement. This suggests that adding more in-context examples has a limited impact when the model has already mastered the task or lacks the inherent capability to solve it.

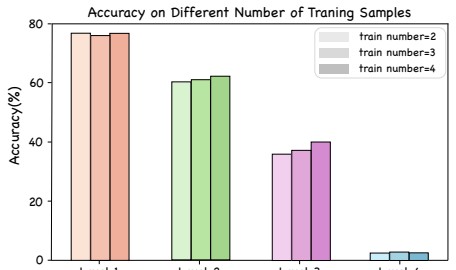

Figure 6: The average accuracy of all evaluated LLMs on different in-context training numbers.

**Impact of the Auxiliary Visual Information.** Previous studies (LeGris et al., 2024; Patterson et al., 2014) have shown that humans tend to perform better on abstract reasoning tasks when the grids are visualized, as visualization can aid in recognizing patterns and rules. Motivated by these findings, we investigate whether adding auxiliary visual information can enhance model performance. Specifically, we visualize each case by two formats: *single-image*, which presents all three training input-output pairs along with the test input in a single image; and *multi-image*, which provides them as seven separate images. What's more, we provide the CoT and non-CoT prompts setting for visual information. Two common CoT instructions are employed: CoT1-"let's think step by step" (Kojima et al., 2022), CoT2-"let's think step by step and reflect on whether your answer is correct" (Shinn et al., 2023). GPT-4o and Claude 3.7 support at least seven images, therefore Table 2 presents their experiment results across all four cognitive levels. Overall, neither adding single-image nor multi-image format inputs can consistently outperform the text-only baseline, and in some instances, accuracy even declines. These results suggest that current models struggle to derive meaningful improvements in abstract reasoning from auxiliary visualized image inputs.

**Impact of the Inference Time.** It is demonstrated in (DeepSeek-AI et al., 2025; OpenAI, 2024b; Qin et al., 2024; Huang et al., 2024) that inference-time scaling plays a crucial role in enhancing model performance on reasoning tasks. Building upon these, we take a step to examine how the

Table 2: Comparison of accuracy across text-only(-), single image (S-Img), multi-image (M-Img), and CoT-augmented settings.

| | GPT-4o | | | | | Claude-3.7 | | | |
|---|---|---|---|---|---|---|---|---|---|
| Vision | L-1 | L-2 | L-3 | L-4 | Vision | L-1 | L-2 | L-3 | L-4 |
| - | **88.42** | 2.86 | 5.00 | 0.0 | - | 95.26 | **25.71** | **45.00** | 15.87 |
| S-Img | 78.95 | 1.44 | 0.00 | 0.00 | S-Img | 96.84 | 17.14 | 31.25 | 15.87 |
| S-Img + CoT1 | 78.87 | 1.17 | 0.00 | 0.00 | S-Img + CoT1 | 96.97 | 16.33 | 31.75 | 15.87 |
| S-Img + CoT2 | 79.11 | 2.86 | 0.00 | 0.00 | S-Img + CoT2 | 95.28 | 18.71 | 33.50 | **16.67** |
| M-Img | 74.74 | 8.57 | 5.00 | 0.00 | M-Img | **97.89** | 17.14 | 35.00 | 12.70 |
| M-Img + CoT1 | 74.82 | **8.82** | 4.50 | 0.00 | M-Img + CoT1 | 96.73 | 17.14 | 34.50 | 14.13 |
| M-Img + CoT2 | 77.29 | 7.86 | **5.50** | 0.00 | M-Img + CoT2 | 97.14 | 16.86 | 35.25 | 13.33 |

model's inference time varies as the complexity of reasoning tasks increases. According to related methods, we use the response latency to measure the inference time. The results are presented in Figure 7. We observe that at the low-level count task, as task complexity increases, the model tends to engage in deeper reasoning and can effectively maintain relatively stable and high accuracy. However, in high-level tasks (i.e., planning), even though the model's inference time increases, it still fails to solve the more complex cases. This indicates that simply increasing inference time is insufficient to compensate for the model's inherent limitations in high-level reasoning.

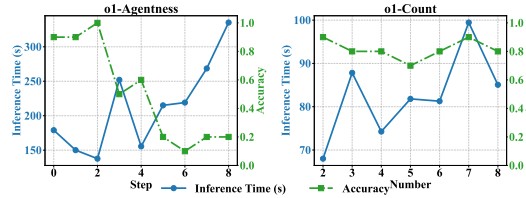

Figure 7: Changing trend in o1's accuracy and inference time as task complexity increases.

Table 3: Results of direction and symmetry.

| | Move | | | | Symmetry | |
|---|---|---|---|---|---|---|
| Model | Up | Down | Left | Right | Horizontal | Vertical |
| DeepSeek-R1 | 91.0 | 94.5 | 88.5 | 85.0 | 48 | 0 |
| o1 | 80.0 | 86.5 | 76.5 | 77.0 | 12 | 8 |
| Claude-3.7 | 82.0 | 95.0 | 48.0 | 44.0 | 52 | 36 |
| o1-mini | 15.0 | 34.0 | 53.5 | 57.5 | 0 | 0 |
| Qwen3-32B | 52.0 | 54.5 | 22.5 | 16.5 | 4 | 0 |
| o3-mini | 7.5 | 20.0 | 34.0 | 38.5 | 0 | 4 |
| QwQ-32b | 28.5 | 17.0 | 35.5 | 32.0 | 12 | 0 |
| SkyWork-OR1-32B | 5.5 | 4.5 | 31.0 | 37.5 | 12 | 0 |
| GPT-4o | 3.0 | 8.5 | 2.0 | 5.5 | 8 | 0 |
| Qwen2.5-32B | 1.0 | 0.0 | 0.0 | 0.0 | 0 | 0 |

## 4.5 CASE STUDY

**Analysis of Spatial Orientations.** Upon closer examination of the results, we find that current models may demonstrate a distinct understanding of spatial orientation compared to humans. As shown in Table 3, the models achieve higher and more consistent accuracy in vertical (up/down) directions than in horizontal (left/right) ones in Move. Similarly, in symmetry tasks, performance is better for horizontal symmetry than for vertical symmetry. However, from the perspective of human cognition, directional distinctions are typically perceived as equivalent (Aflalo & Graziano, 2008; Ambinder et al., 2009). These findings suggest that current LLMs may exhibit systematic divergences from human cognitive patterns in processing spatial orientation.

**Analysis of Error Cases.** As shown in Figure 8, we randomly select error cases from four cognitive levels and visualize the model output alongside the corresponding ground-truth for analysis. In Level-1 and Level-2, the differences between the model's error predictions and the correct answers are relatively subtle, indicating that the model roughly understands the required operation. However, in Levels-3 and Level-4, the incorrect outputs become significantly more disorganized and divergent from the ground truth, suggesting a complete failure to grasp the underlying rule. This is especially evident in Level-4, where physical concepts pose substantial challenges to the models. These observations highlight that as the cognitive level increases, the nature of model errors becomes increasingly complex and unreasonable. The results of two auxiliary evaluation metrics: grid size precision and grid matching percentage in Appendix E.2 also confirm this circumstance. More detailed analysis of model failure modes on high-level tasks, see the Appendix Figure 16

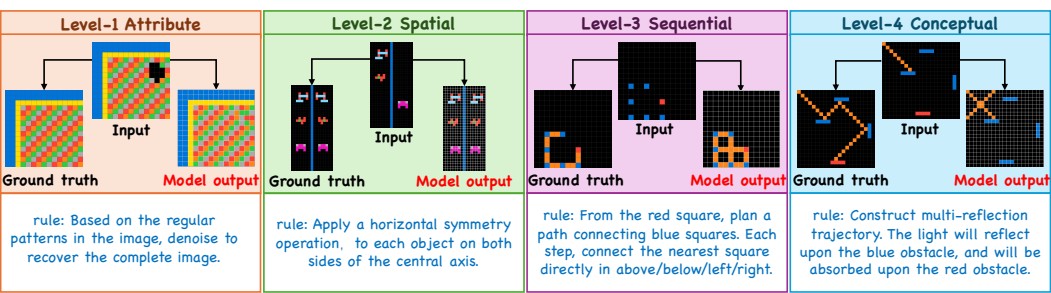

Figure 8: Error cases on o1: input, ground truth, and model output grids are visualized for each case.

## 5 CONCLUSION

In this work, we present DRE-Bench, a benchmark designed to evaluate the fluid intelligence of large language models (LLMs) through abstract reasoning tasks. By combining a hierarchical task design, a scalable generator–solver pipeline, and dynamic task instantiation, DRE-Bench provides interpretability, scalability and robustness beyond prior benchmarks. Our experiments show that while reasoning-oriented models outperform general LLMs, their accuracy declines as cognitive level increases and case complexity rises. The results indicates that true fluid intelligence remains out of reach for current LLMs. DRE-Bench offers a principled framework for tracking reasoning progress and guide the development of future models with stronger generalizable intelligence.

### ETHICS STATEMENT

This work complies fully with the ICLR Code of Ethics. No private, sensitive, or personally identifiable information was collected or used. The study involves no human subjects, no experiments on vulnerable populations, and no interventions requiring IRB approval. We confirm that our methodology and results do not raise foreseeable risks of harm, misuse, or ethical concerns beyond standard scientific research practices.

### REPRODUCIBILITY STATEMENT

We present DRE-Bench, a benchmark for evaluating the fluid intelligence of large language models via abstract reasoning tasks structured in a four-level cognitive hierarchy. Compared with previous benchmarks, DRE-Bench probes latent rules across tasks and variants to provide interpretability, dynamic robustness, and scalability for tracking reasoning capabilities. We affirm the value of reproducibility in scientific research and therefore summarize the details of dataset, method, and experiments as follows:

- Dataset. The detailed document and distribution of DRE-Bench are in Appendix C. And our dataset and all pairs of generator and solver have been available at the anonymous github link `https://anonymous.4open.science/status/DRE-Bench-8098`;

- Method. The prompt templates to instruct code agent are detailed in Appendix D;

- Experiment. Details about evaluated LLMs, results of two auxiliary evaluation metrics, more dynamic evaluation curves, example of two visual formats, and detailed table of variance are in Appendix E;

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

# A    APPENDIX

# B    LLM USAGE STATEMENT

We used LLMs(Gpt-5) to refine the writing, including checking grammar, polishing, and correcting typos. To ensure the writing quality, we further check and refine all the LLMs generated text. We assure that ideas, methods, code implementations, experiments, analyses, and conclusions aredone by human researchers ourselves.

# C    DETAILS OF DRE-BENCH

## C.1    DETAILED DATASET CONTENT AND DISTRIBUTION

To provide a more concrete overview of our dataset, we present its detailed composition and distribution in the table 4 below. This includes the specific rules, tasks, and descriptions across the four cognitive levels, along with the corresponding variables, variable ranges, and the number of data samples for each task.

Table 4: Descriptions, cognitive levels, variables, value ranges, and examples of the six atomic operations used in this paper.

| Level | Name | Description | Variable | Value Range | Number |
|-------|------|-------------|----------|-------------|--------|
| Attribute | Size | Change the size of the whole grid or one object while maintaining the rules. | size | {10–30} | 629 |
| | Count | Change the number of grids to be counted. | number | {2–10} | 570 |
| | Shape | Change the shape of an object. | shape | {1–10} | 450 |
| Spatial | Moving | Move the object several steps towards one of {Up, Down, Left, Right, Up Right}. | distance | {1–30} | 1500 |
| | Rotation | Rotate the object around the {Endpoint, Center}. | angle | {0°, 360°} | 108 |
| | Symmetry | Perform {Vertical, Horizontal, Center} symmetry of the object. | number | {1, 9} | 75 |
| Sequential | Categorization | Classify objects based on examples, and apply the corresponding rule to each category. | category | {1, 6} | 65 |
| | Sort | Rearrange objects according to a sequential rule. | order | {1, 9} | 240 |
| | Planning | Start from an object, plan and execute a path. | step | {1, 9} | 105 |
| Conceptual | Gravity | Objects in mid-air should fall downward according to gravity. | number | {1, 9} | 63 |
| | Reflection | The light reflects upon hitting walls. | number | {1, 9} | 100 |
| | Expansion | Objects expand when heated until obstructed. | number | {1–9} | 50 |
| Total | – | – | – | – | 3955 |

## C.2    DATASET DOCUMENT

We provide comprehensive documentation of our dataset along with its intended use cases. The dataset and accompanying resources are available at the following link: https://anonymous.4open.science/status/DRE-Bench-8098, which includes metadata, format details, and so on.

## D DETAILS OF METHOD

Our method employs two sequential system prompts to instruct code agent to implement generator and solver functions for each rule task. Based on the designed rule and corresponding constraints, the first system prompt guides the LLM to generate a structured code-like rule description, And the second system prompt translates this description into a complete Pygame program. We tested with different LLM-based code agents, including Gemini 2.5-Pro, Claude Opus 4-thinking, GPT-o3, and GPT-4o, and ultimately selected Gemini-2.5-Pro as the code agent in our experiments due to its higher success rate of generation. This two-prompt design ensures a clear division between rule modeling and executable code generation.

**System Prompt 1: Rewrite the given rule and constraints into a structured rule description**

```
"""
You are an imaginative world architect and a technical artist. Your mission is to fuse a
    series of fundamental latent rules provided by the user (e.g., physics, math, artistic
    concepts) to create a concrete, detailed, and dynamic virtual scene.

Your output must adhere to the following guidelines:
1. **Structured Output**: Use a clear key-value format to describe the scene, making it
    easy to parse later.
2. **Code-like Description**: Use precise, quantifiable language, as if writing pseudocode
    or a configuration file. Avoid vague, literary descriptions.

3. **Dynamics and Interaction**: Focus on describing the behavior of elements, their
    interaction rules, and how they embody the user's core rules.

Example Output Format:
Scene Name: [A creative name for the scene]
Core Rules: [Summarize the user's concepts and how they are manifested in the scene]
Element List:
 - Element A:
   - Type: [e.g., Static Body, Dynamic Particle, Interactive Character]
   - Visual Description: [A concise description of its appearance, material, color]
   - Initial State: [Position coordinates, rotation angle, initial velocity, etc.]
   - Behavioral Rules: [Describe how it moves, changes, and embodies the core concepts]
 - Element B:
   ...
Physics & Interaction Rules:
 - Rule 1: [e.g., Global gravity is set to a vector of (0, 0.1)]
 - Rule 2: [e.g., When Element A and B collide, trigger a 'symmetrical' bounce effect]
 - Rule 3: [e.g., An element must find a path from a start to an end point, demonstrating '
    pathfinding']
"""
```

**System Prompt 2: Instruct the code agent to produce generator and solver fuctions based on the detailed rule description.**

```
"""
You are a senior Python game developer and an expert in using the Pygame library. Your task
    is to write a single, complete, and executable Pygame program that simulates the scene
    , strictly following the structured scene description provided by the user.

Your code must adhere to the following guidelines:
1. **Code Completeness**: Generate a single, complete Python script that includes all
    necessary Pygame initialization, the main loop, event handling, and rendering code.
2. **Precise Implementation**: The code's logic must accurately implement every element,
    behavior, and physical rule from the scene description.
3. **Readability**: The code must be clean and well-commented. Especially in the parts
    implementing core concepts (like gravity, pathfinding, rotation), explain how the code
    corresponds to the design document.
4. **No External Assets**: Use Pygame's drawing functions (e.g., `pygame.draw`) to create
    geometric shapes. Do not rely on any external image or audio files.
"""
```

## E EXPERIMENTAL DETAILS

### E.1 DETAILS OF EVALUATED LLMS

Table 6 lists the 11 representative LLMs examined in this study. To facilitate transparent comparison, each model is annotated along four dimensions: Model Type (General models are trained for broad-domain language generation, whereas Reasoning models have undergone additional fine-tuning or

alignment specifically targeting reasoning tasks.), Param (Whenever the developer discloses the parameter count, we report it verbatim. For proprietary APIs that do not reveal their scale, the entry is marked " — ".), Vision Modality, and Open-source.

## E.2 RESULTS OF TWO AUXILIARY EVALUATION METRICS ON DRE-BENCH

To evaluate more thoroughly, we have provided the results of LLMs by their accuracy, the variance of accuracy, and the accuracy curve. Besides, we further calculate two additional metrics to further assess the model's performance:

Grid Size Precision: checks if the LLM's output grid size matches the ground truth (GT) grid. If matching scores 1; otherwise, it scores 0. This assesses the model's ability to handle grid dimensions.

Grid Matching Percentage: the proportion of matching elements between the response and GT grids. If the grid sizes are unequal, the score is set to 0. This percentage offers a finer-grained score.

Table 5: The average results of grid size precision/grid matching percentage/original accuracy in four levels.

| Model | Level 1 Attribute | | | | Level 2 Spatial | | | | Level 3 Sequential | | | | Level 4 Conceptual | | | |
|---|---|---|---|---|---|---|---|---|---|---|---|---|---|---|---|---|
| | Size | Count | Shape | Avg-1 | Rotation | Move | Symmetry | Avg-2 | Category | Sort | Planning | Avg-3 | Optics | Mechanics | Thermal | Avg-4 |
| *General LLMs* | | | | | | | | | | | | | | | | |
| Claude-3.7 | 100/99/65 | 100/91/63 | 100/42/13 | 100/83/58 | 100/88/68 | 99/64/57 | **100/89/49** | 99/78/58 | **100/73/54** | 100/94/2 | **100/88/54** | **100/83/44** | **100/61/8** | 100/75/15 | 100/59/0 | 100/65/7 |
| Qwen3-32B | 91/90/61 | **100/95/71** | 100/45/18 | 96/82/60 | 100/67/51 | 90/42/29 | 36/20/1 | 77/43/27 | 85/57/7 | 83/77/3 | 100/63/8 | 88/64/7 | 64/22/0 | 100/50/0 | 100/51/0 | 88/41/0 |
| GPT-4o | 100/89/62 | 100/84/44 | 100/40/13 | 100/76/51 | 99/59/27 | 95/10/3 | 86/65/2 | 93/40/9 | 98/66/8 | 100/95/2 | 99/64/8 | 98/73/7 | 96/47/0 | 100/59/0 | 98/40/0 | 98/49/0 |
| Qwen2.5-32B | 72/61/44 | 100/78/28 | 100/29/6 | 89/60/35 | 67/18/5 | 17/1/0 | 5/3/0 | 28/6/1 | 91/54/4 | 63/58/1 | 93/38/7 | 84/51/4 | 96/42/0 | 93/34/0 | 66/33/0 | 85/36/0 |
| *Reasoning LLMs* | | | | | | | | | | | | | | | | |
| o1 | 99/97/64 | 100/88/60 | 100/65/58 | 99/86/62 | 100/97/93 | 94/76/69 | 64/53/6 | 87/75/58 | 87/71/26 | 100/94/11 | 100/86/53 | 94/81/28 | 96/52/0 | 100/60/7 | 100/62/0 | 98/58/2 |
| DeepSeek-R1 | 99/99/60 | 100/95/69 | 100/24/8 | 99/80/57 | 100/89/82 | **95/85/78** | 92/81/16 | **92/81/62** | 100/89/44 | 100/90/0 | 100/86/44 | 100/89/35 | 100/57/0 | 100/53/1 | 100/58/0 | 100/56/0 |
| o1-mini | 85/83/40 | 100/93/65 | 100/43/18 | 94/78/46 | 90/69/63 | 63/36/32 | 17/10/0 | 57/38/31 | 70/56/43 | 76/72/7 | 97/74/43 | 79/65/36 | 76/29/0 | 22/9/0 | 80/47/0 | 59/28/0 |
| o3-mini | 78/71/31 | 99/92/60 | **100/78/71** | 91/81/45 | 82/56/50 | 55/23/20 | 21/14/1 | 53/30/23 | 54/42/25 | 78/74/7 | 91/47/25 | 71/52/21 | 76/36/0 | **100/73/31** | 73/42/0 | **83/50/10** |
| QwQ-32B | **94/94/78** | 100/95/61 | 100/35/13 | **97/81/65** | 100/82/64 | 85/34/22 | 88/64/4 | 90/57/29 | 88/62/12 | 92/86/0 | 100/79/34 | 92/73/14 | 100/44/0 | 93/37/0 | 82/31/0 | 91/37/0 |
| SkyWork-OR1-32B | 93/92/59 | 100/95/68 | 100/43/13 | 97/82/57 | 100/85/64 | 64/27/15 | 94/71/4 | 83/57/25 | 96/62/9 | 100/92/0 | 100/80/36 | 98/75/12 | 100/44/0 | 96/43/0 | 2/0/0 | 66/29/0 |

As Table 5, most models have high grid size precision, indicating they can roughly infer the overall size of the required output grid. Meanwhile, grid matching percentages are lower, but remain above binary accuracy, suggesting that models often produce outputs close to the ground truth. And both grid size precision and grid matching percentage decrease as cognitive level increases, consistent with the original accuracy, validating our data framework.

Table 6: Evaluated LLMs in this study with type, specification, vision modality, and open-source status

| Model Name | Model Type | Param | Vision Modality | Open-source |
|---|---|---|---|---|
| Claude-3.7 | General | – | Multi-modal | No |
| Qwen3-32B | General | 32B | Text-only | Yes |
| GPT-4o | General | – | Multi-modal | No |
| Qwen2.5-32B | General | 32B | Text-only | Yes |
| o1 | Reasoning | – | Multi-modal | No |
| DeepSeek-R1 | Reasoning | 671B | Text-only | Yes |
| o1-mini | Reasoning | – | Text-only | No |
| o3-mini | Reasoning | – | Text-only(API) | No |
| QwQ-32B | Reasoning | 32B | Text-only | Yes |
| SkyWork-OR1 | Reasoning | 32B | Text-only | Yes |

## E.3 MORE DYNAMIC EVALUATION CURVES

Since our generator is capable of producing data with varying levels of complexity, we conduct a fine-grained evaluation to assess model performance across different cognitive levels. The four figures below illustrate performance curves of all rules corresponding to each cognitive level.

In the rules in Level-1, namely size, count, and shape, the models achieved relatively high average accuracy and stable performance since these tasks involve basic enumeration without substantial cognitive demands.

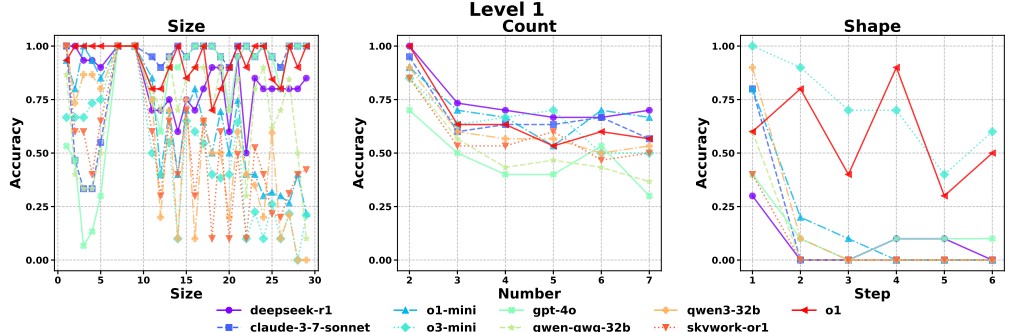

Figure 9: Model performance curves under varying complexities in level-1.

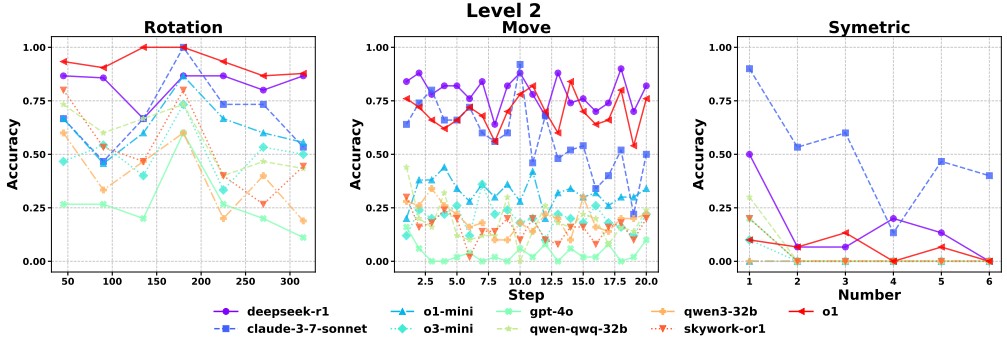

Figure 10: Model performance curves under varying complexities in level-2.

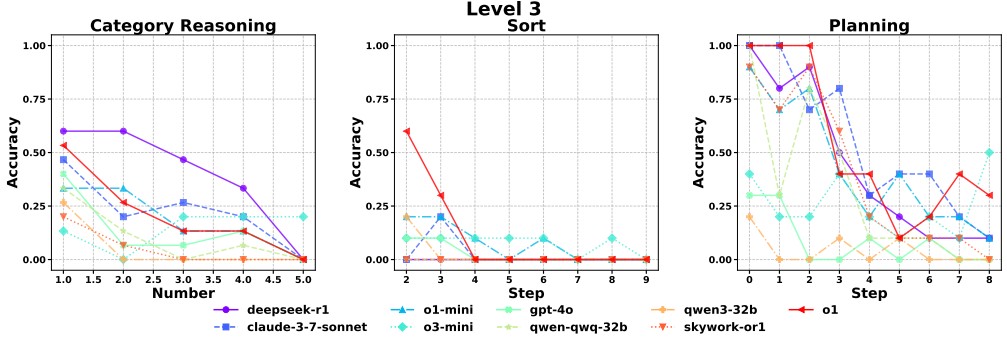

Figure 11: Model performance curves under varying complexities in level-3.

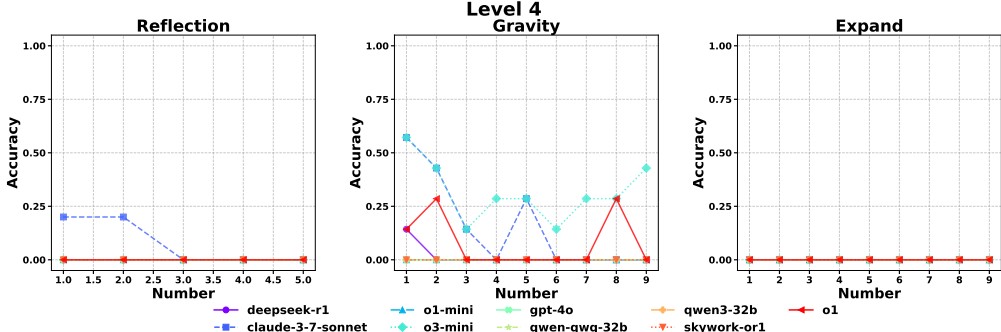

Figure 12: Model performance curves under varying complexities in level-4.

Figure 13: Samples of visualization format to multimodal LLMs.

For the rotation, move, and symmetry rules in Level-2, performance gaps between models became more obvious compared to Level-1. But models still remain stable in these rules, and haven't dropped much.

Regarding tasks in Level-3, we observe a substantial performance drop as the complexity of rules increases, whether on category reasoning, sorting, or planning.

In Level-4, although the complexity of the cases is small, models still fail to provide correct solutions and consistently present low accuracy.

### E.4 VISUALIZATION FORMAT

To provide multimodal LLMs with visual information, we designed two methods for incorporating the visual modality: one using a single image, and the other using multiple images. The figure 13 below shows some examples of single-image visual information. And multi-image means giving six input and output images of training samples and one input image of a testing sample to the LLM, respectively, and telling it what these images represent.

### E.5 DETAILED TABLE

Since plotting the accuracy and variance of all models together would make the graph unclear (or: cluttered), the Table 7 lists the specific accuracy and variance for each model to supplement the scatter plot in the main text.

Table 7: Detailed model performance across reasoning tasks (Accuracy [%] / Variance)

| Model | Level 1: Attribute | | | Level 2: Spatial | | | Level 3: Sequential | | | Level 4: Conceptual | | |
|---|---|---|---|---|---|---|---|---|---|---|---|---|
| | Size | Number | Shape | Rotation | Move | Symmetry | Category | Sort | Planning | Optics | Mechanics | Thermal |
| o1-mini | 69.48/0.0133 | 65.43/0.0058 | 18.33/0.0814 | 63.04/0.0336 | 32.10/0.0215 | 0.00/0.0 | 43.33/0.0154 | 7.50/0.0069 | 43.33/0.0778 | 0/0.0 | 0.00/0.0 | 0/0.0 |
| o3-mini | 55.37/0.0131 | 60.10/0.0145 | 71.67/0.0381 | 50.14/0.0471 | 20.00/0.0173 | 1.33/0.0021 | 25.56/0.0183 | 7.50/0.0019 | 25.56/0.0180 | 0/0.0 | 31.75/0.0 | 0/0.0 |
| gpt-4o | 35.20/0.0271 | 44.48/0.0209 | 13.33/0.0156 | 27.30/0.0328 | 3.80/0.0082 | 2.67/0.0085 | 8.89/0.0354 | 2.50/0.0019 | 8.89/0.0143 | 0/0.0 | 0.00/0.0 | 0/0.0 |
| Claude-3.7 | 50.48/0.0232 | 63.14/0.0037 | 13.33/0.0889 | 68.57/0.0599 | 57.80/0.0606 | 49.33/0.0853 | 54.44/0.0392 | 2.50/0.0044 | 54.44/0.1025 | 8/0.2 | 15.87/0.3 | 0/0.0 |
| deepseek-r1 | 76.92/0.0074 | 69.43/0.0015 | 8.33/0.0114 | 82.72/0.0085 | 78.90/0.0159 | 16.00/0.0299 | 44.44/0.0169 | 0.00/0.0 | 44.44/0.1202 | 0/0.0 | 1.59/0.1 | 0/0.0 |
| o1 | 80.79/0.0106 | 60.00/0.0063 | 58.33/0.0447 | 93.08/0.0101 | 69.69/0.0275 | 6.67/0.0064 | 26.67/0.0415 | 11.25/0.0436 | 53.33/0.1178 | 0/0.0 | 7.94/0.0 | 0/0.0 |
| qwq-32b | 78.59/0.0574 | 61.05/0.0190 | 13.33/0.0889 | 64.76/0.0440 | 22.80/0.0295 | 4.00/0.0192 | 12.31/0.0430 | 0.00/0.0 | 34.44/0.1247 | 0/0.0 | 0.00/0.0 | 0/0.0 |
| skywork-32b | 59.62/0.0405 | 68.95/0.0110 | 13.33/0.0456 | 64.76/0.0740 | 15.90/0.0167 | 4.00/0.0192 | 9.23/0.0340 | 0.00/0.0 | 36.67/0.0844 | 0/0.0 | 0.00/0.0 | 0/0.0 |
| qwen3-32b | 61.79/0.0574 | 71.05/0.0070 | 18.33/0.1347 | 51.43/0.0790 | 29.20/0.0353 | 1.33/0.0021 | 7.69/0.0580 | 3.75/0.0100 | 8.89/0.0099 | 0/0.0 | 0.00/0.0 | 0/0.0 |
| qwen2.5-32b | 44.72/0.1156 | 28.42/0.0260 | 6.67/0.0122 | 5.71/0.0270 | 0.20/0.0002 | 0.00/0.0 | 4.62/0.0210 | 1.25/0.0010 | 7.78/0.0062 | 0/0.0 | 0.00/0.0 | 0/0.0 |

### E.6 SCATTER PLOTS OF ACCURACY VERSUS VARIANCE ON ALL MODELS

In Figure 5, we select several top-performing models and present their results. We now additionally present the full results of all models in Figure 14, and we are still able to draw similar conclusions to the main content.

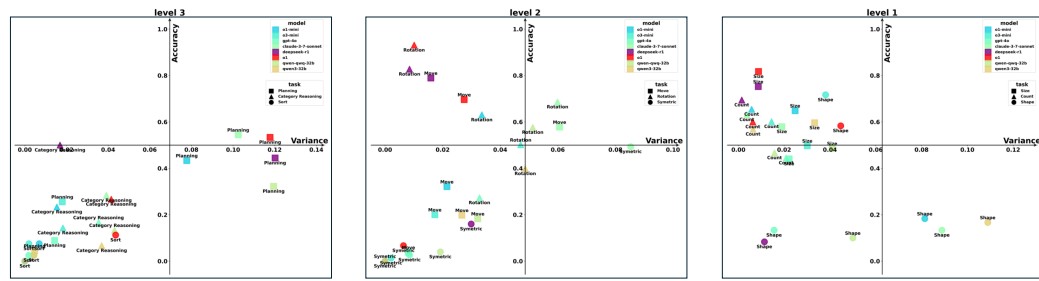

Figure 14: Scatter plots of accuracy versus variance on all models across four levels.

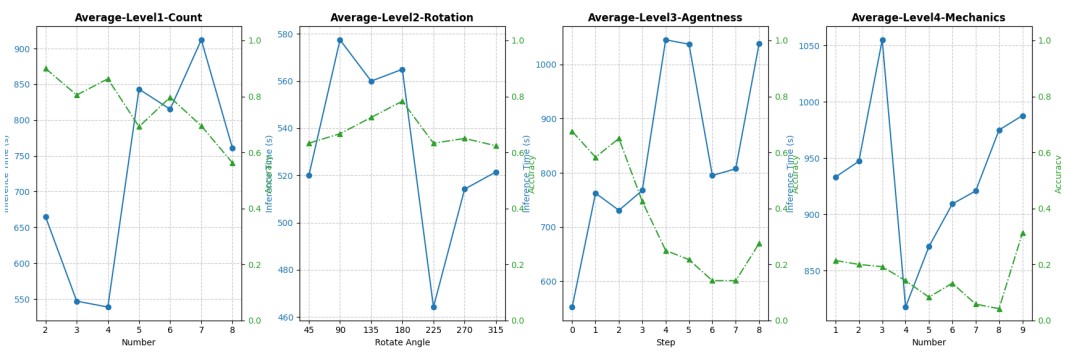

Figure 15: Results of inference time scaling in more models and more tasks.

### E.7 MORE EXPERIMENTS OF INFERENCE TIME SCALING

To supplement Figure 7, we not only evaluated inference time on a larger set of models but also on more tasks across four levels. To obviously reveal the trend of accuracy changing, we selected the task with the highest average model accuracy for each level—namely, count, rotation, agent, and mechanics. In addition, the Level-4 mechanics task is so challenging that only Claude 3.7, O1, and O3-mini achieve non-near-zero accuracy. Thus, we expanded the inference-time experiments on the four tasks using these three models.

As Figure 15, in the level-1 count task, as task complexity increases, the model tends to engage in deeper reasoning and can effectively maintain relatively stable and high accuracy. In the Level-2 rotation task, the models exhibit a clear trend: rotations by single multiples of 45° yield substantially lower accuracy than rotations by multiples of 90°. This is reasonable, as prior studies (Appelle, 1972) have demonstrated that it is more difficult for humans to perceive lines at 45° diagonal orientations. However, in high-level tasks (i.e., level-3,level-4), even though the model's inference time increases, it still fails to solve the more complex cases. This indicates that simply increasing inference time is insufficient to compensate for the model's inherent limitations in high-level reasoning.

### E.8 DETAILED ANALYSIS OF MODEL FAILURE MODES

In detail, we first visualized the model outputs for the Level-3 and Level-4 tasks and found that the error patterns are highly diverse. Next, we compare errors made by different models on the same task, and summarize some common error patterns from the Category, Planning, Gravity, and Reflection tasks, as illustrated in Figure 16:

**Category Task:** In this task, most models were able to correctly identify the object blocks, but failed to fill in the correct colors. A common issue was misunderstanding the color-to-category correspondence (error 1, 2, 3). Another issue was missing objects during the recoloring process,

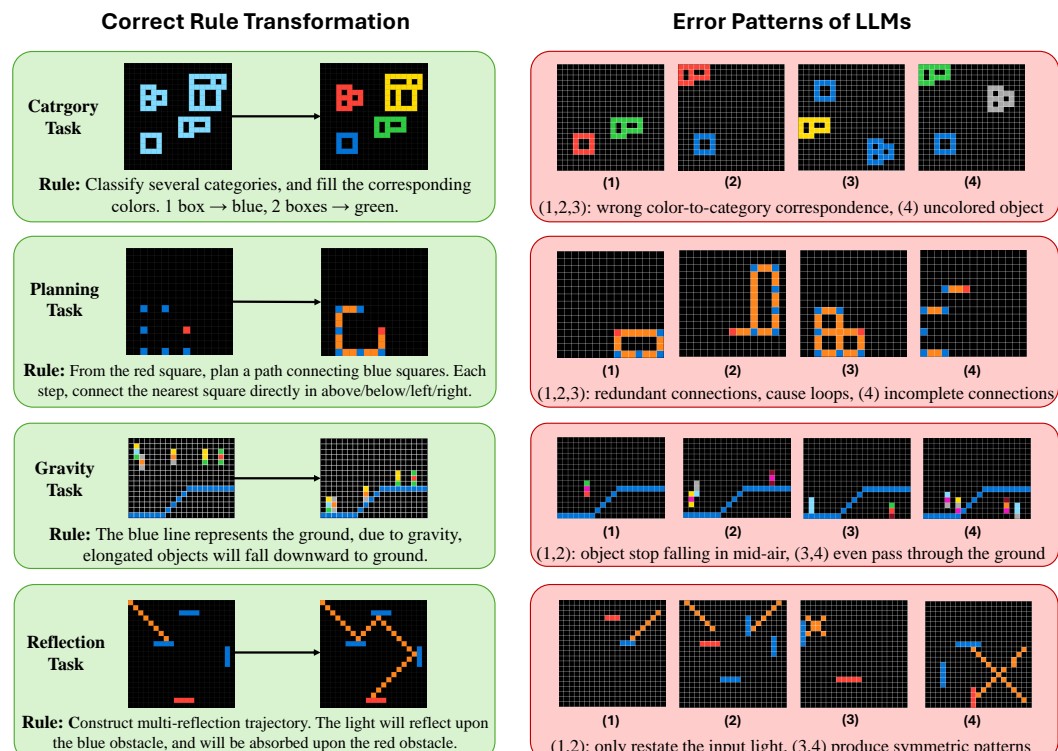

Figure 16: Some typical error patterns in high-level tasks: Category, Planning, Gravity, Reflection

where some objects were left uncolored (error 4). However, most models were able to complete part of the coloring task, indicating that they correctly identified some of the object categories.

**Planning Task:** Here, many models struggled with the required path-planning ability. A large portion of cases produced redundant connections, causing the wrong loops instead of a directed path (error 1, 2, 3). Conversely, others produced incomplete connections, only managed to construct partial paths(error 4). This may be because the planning task involves multiple steps, which creates substantial difficulty for models.

**Gravity Task:** In this task, several cases captured the basic physical sense that objects should move downward, but how objects interact with the ground was captured incompletely. Some cases caused the object to inadequate-descend (error 1, 2, stopping in mid-air), whereas others caused it to over-descend (error 3, 4, passing through the ground), showing that they failed to correctly infer the concept of gravity.

**Reflection Task:** Most cases did not get the rule that a light ray should reflect upon hitting obstacles. Some outputs simply restated the input pattern (error 1, 2), while others produced symmetric patterns that is irrelevant to the intended reflection (error 3, 4). Generally, models perform poorly on this task, and compared with gravity, they show an even weaker understanding of the physical concept of reflection.

## E.9 MORE DETAILED DESIGN AND STATISTICAL ANALYSIS OF HUMAN EXPERIMENT

### E.9.1 DETAILED AGE DISTRIBUTION

We further analyze the age distribution and geographical distribution of participants in the human study, and the results are presented in Figure 17. We can see that the age distribution of our participants is relatively balanced, embodying a certain degree of representativeness.

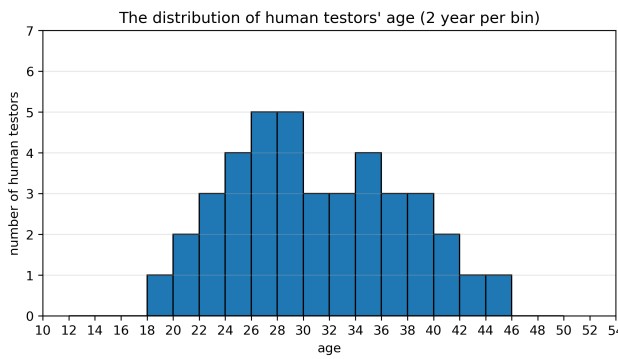

Figure 17: The distribution of human participants' age (2 years per bin).

### E.9.2 SIGNIFICANCE TESTING

To perform significance testing, we calculate the standard deviation of human accuracy on different tasks and conduct an independent t-test between the two distributions of human and model accuracy.

**The standard deviation of human accuracy:** The results of Table 8 show the standard deviation among human participants in different tasks. Most standard deviations belong to a normal range (less than 3), which can validate the credibility of our human study.

Table 8: The mean and standard deviation among human participants in different tasks.

| Result | Level 1: Attribute | | | Level 2: Spatial | | | Level 3: Sequential | | | Level 4: Conceptual | | |
|---|---|---|---|---|---|---|---|---|---|---|---|---|
| | Size | Count | Shape | Rotation | Move | Symmetry | Category | Sort | Planning | Optics | Mechanics | Thermal |
| Mean | 75.96 | 82.02 | 71.72 | 84.65 | 77.78 | 44.25 | 75.75 | 29.49 | 89.90 | 49.70 | 76.16 | 16.16 |
| Std | 0.93 | 1.42 | 1.61 | 1.67 | 1.64 | 1.34 | 1.70 | 1.57 | 1.52 | 1.52 | 1.61 | 1.23 |

**Independent t-test between the two distributions of human and model accuracy:** Furthermore, based on the evaluation results of models and humans, we conducted an independent t-test on their distributions. The p-values for all tasks are reported in Table 9, where the majority of values are below 0.05. These results demonstrate the statistical significance of humans' and models' results on our proposed benchmark, validating that the claimed conclusion—humans relatively outperform current models—is well supported.

Table 9: Independent t-test between the two distributions of human and model accuracy. (t-statistic / p-value)

| Result | Level 1: Attribute | | | Level 2: Spatial | | | Level 3: Sequential | | | Level 4: Conceptual | | |
|---|---|---|---|---|---|---|---|---|---|---|---|---|
| | Size | Count | Shape | Rotation | Move | Symmetry | Category | Sort | Planning | Optics | Mechanics | Thermal |
| t-statistic | -3.7147 | -1.1487 | -3.0798 | -1.7772 | -3.8474 | -2.9810 | -2.2172 | -2.2668 | -5.0581 | -3.6903 | -9.9144 | -2.8960 |
| p-value | 0.0014 | 0.2649 | 0.0061 | 0.0915 | 0.0010 | 0.0076 | 0.0390 | 0.0352 | 6.9822e-05 | 0.0015 | 6.0433e-09 | 0.0092 |
| p-value< 0.05 | ✓ | | ✓ | | ✓ | ✓ | ✓ | ✓ | ✓ | ✓ | ✓ | ✓ |

### E.9.3 THE AVERAGE SOLVING TIME PER TASK

Specifically, based on the dwell time on each question in the questionnaire, we computed the average solving time for all participants across each task. Table 10 presents the average solving time and accuracy of human participants for tasks on different levels. We can observe that, despite some fluctuations, the average human solving time generally increases as the task level becomes higher. This could further substantiate our claimed cognitive alignment and reveal the underlying difficulty

gradient. However, we also note that for tasks with lower accuracy, the time spent by participants does not decrease too much. This indicates that participants engaged in continuous reasoning and attempts, rather than giving up quickly when the tasks were challenging, further substantiating our human study.

Table 10: Average accuracy and solving time of human participants across cognitive levels. (seconds)

| Metric | Level 1: Attribute | | | Level 2: Spatial | | | Level 3: Sequential | | | Level 4: Conceptual | | |
|---|---|---|---|---|---|---|---|---|---|---|---|---|
| | Size | Count | Shape | Rotation | Move | Symmetry | Category | Sort | Planning | Optics | Mechanics | Thermal |
| Accuracy-avg | 75.56 | 82.22 | 68.89 | 91.11 | 75.56 | 46.67 | 73.33 | 24.44 | 88.89 | 46.67 | 77.78 | 17.78 |
| Time-avg | 73.4 | 63.4 | 62.2 | 90.2 | 88.8 | 93.0 | 109.8 | 106.4 | 105.8 | 165.8 | 132.1 | 147.3 |

### E.9.4 DETAILED INSTRUCTIONS OF THE QUESTIONNAIRE

We provided very clear instructions in the questionnaire README, which explained the definition of abstract reasoning tasks, identified the given training samples, and explained how to fill in the answers for the test samples. Such explicit and detailed instructions improve the usability of the questionnaire and ensure the rigor of the human study. The specific instruction is as follows:

**Detailed instructions of the questionnaire.**

```
"""
Your task is to complete a questionnaire consisting of 400 questions. Each question is an
    abstract reasoning task. To solve each question, you can follow the steps below:

1, Reason out the transformation rule by analyzing the three training sample pairs, namely
    how the input grids are converted into corresponding output grids.

2, Apply the transformation rule to the test input grid and generate the correct test
    output grid.

For convenience, the test output grid has been pre-initialized with the content of the test
     input grid, so you can directly perform the rule-based transformation.

When you finish a question, click "Next" to proceed to the next one. After completing all
    questions, click "Final Submit."
"""
```

### E.9.5 THE UI INTERFACE DESIGN OF THE RELEASED QUESTIONNAIRE

We carefully designed the UI interface for the questionnaire and have open-sourced the UI code at https://anonymous.4open.science/r/DRE-Bench-8098. To exclude potential influencing factors in the human study, we adopted several user-friendly measures in the questionnaire design, mainly as follows:

**General interface.** As the ui example in Figure 18, the first three rows correspond to the training sample pairs, where the input grid is shown on the left and the output grid on the right. In the fourth row, the left side displays the test input grid, while the right side is the test output grid that participants must fill in.

**To mitigate visual overload caused by large grids.** Many of the abstract reasoning tasks involve relatively large grids, often exceeding 15×15 in size. As previous research (Baddeley, 2012), humans could easily lose track of numerical positions within the grid, causing errors even when they correctly understood the rule. In contrast, LLMs are less affected due to their strong ability to restate longer contexts (Hsieh et al., 2024; Bai et al., 2024). To fill this gap, we added row and column numbers to the test input grid in the UI, making it easier for humans to locate specific positions (as shown in the red box in Figure 19).

**To reduce the unnecessary burden of reproducing large grids.** In abstract reasoning tasks, the ground truth output is derived from applying a transformation rule on the test input. Therefore, in many tasks, a large portion of the output grid remains identical to the input. Since the restate

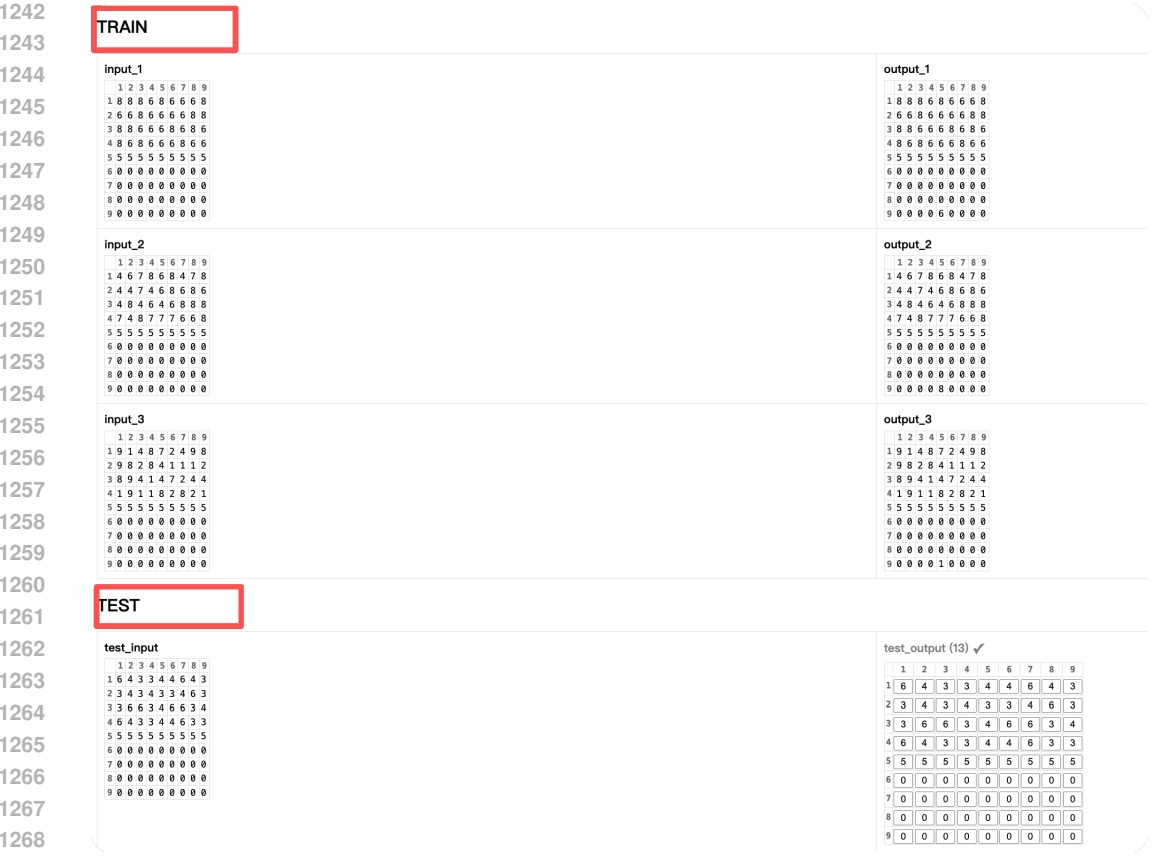

Figure 18: The general UI interface of an example of abstract reasoning question

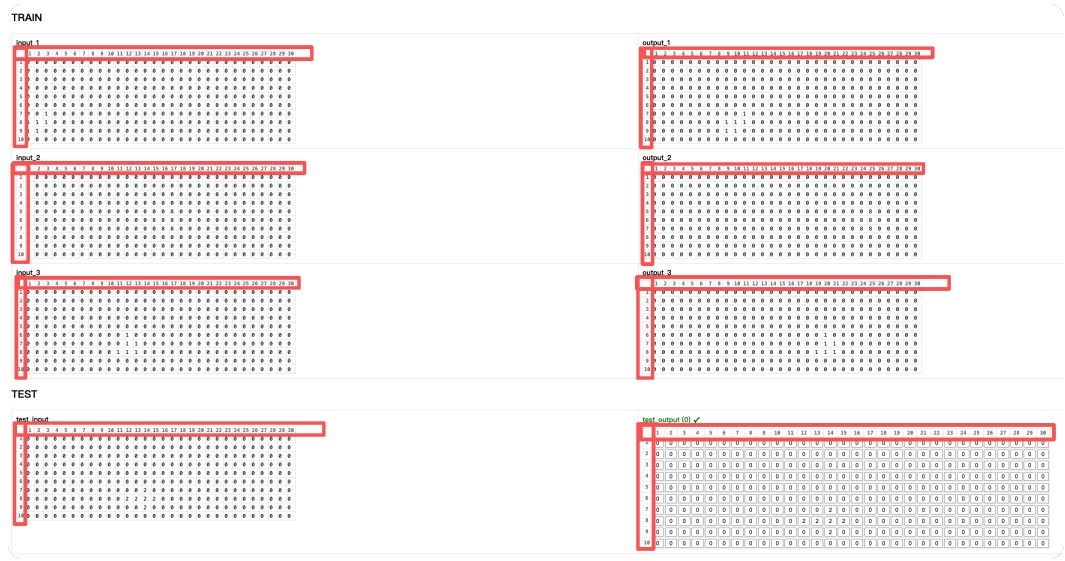

Figure 19: The marked row and column numbers of grids.

ability of models is stronger than humans (Hsieh et al., 2024; Bai et al., 2024), we initialized the output grid with the input grid, helping humans to focus primarily on the reasoning logic and rule transformation. (as shown in the red box in Figure 20).

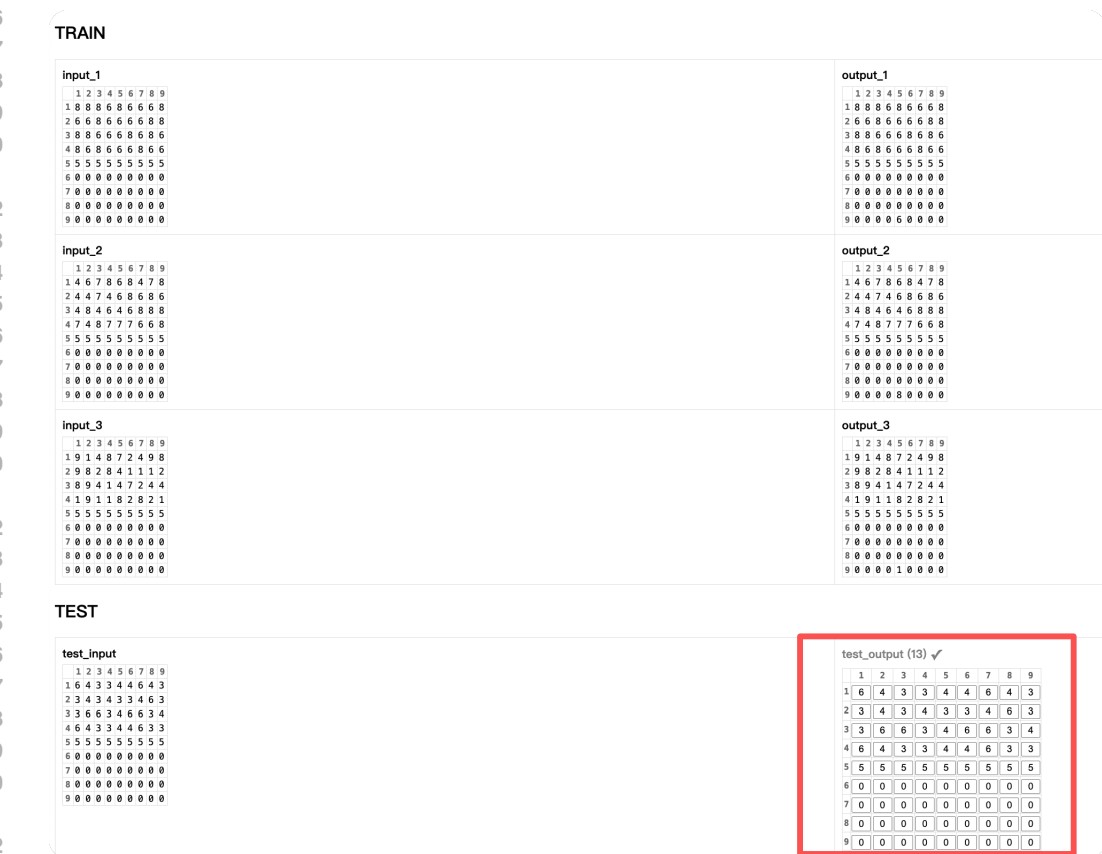

Figure 20: The output grid is initialized with the input grid, helping humans to focus primarily on the reasoning logic and rule transformation.

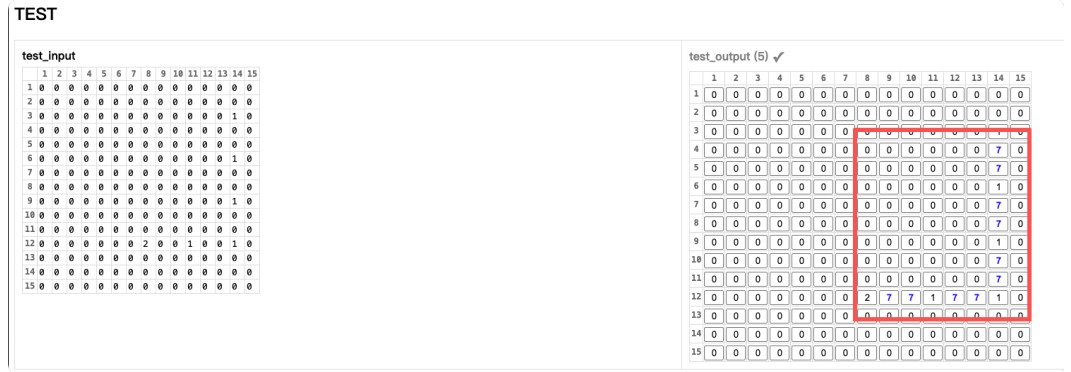

Figure 21: The highlighted cells in the output grid that differ from the input grid in bright blue.

**To highlight the actual reasoning operations.** To further emphasize the reasoning logic, we visually highlighted all cells in the output grid that differ from the input grid in bright blue. This provides a user-friendly and intuitive way to visualize the transformation human induced. (as shown in the red box in Figure 21).

In summary, the final questionnaire version has undergone multiple rounds of review by the co-authors. To exclude potential confounding factors, we have carefully designed UI enhancements to make the questionnaire more accessible to human participants.

