# OpenReview forum: "Truly Assessing Fluid Intelligence of Large Language Models through Dynamic Reasoning Evaluation"
_ICLR.cc/2026/Conference — Submitted to ICLR 2026_

### Official Review · Reviewer_K9CV · 2025-10-27

**Soundness:** 3
**Presentation:** 3
**Contribution:** 3
**Rating:** 6
**Confidence:** 3

**Summary:**

This paper proposes DRE-Bench, a cognitive-level benchmark for abstract reasoning, showing that while reasoning-focused LLMs outperform general ones, all models still fail on higher-level sequential and conceptual tasks, revealing a significant gap from human fluid intelligence.

**Strengths:**

**Cognitive alignment:** Tasks are designed based on the psychological cognitive hierarchy (attribute, spatial, sequential, and conceptual) to structuredly evaluate the abstract reasoning ability of LLMs. This structured assessment is reasonable and meaningful.

**Dynamic robustness:** Generating diverse tasks by varying parameters to avoid data leakage while revealing the stability and generalization ability of the model under different levels of complexity, which is novel.

**Weaknesses:**

This paper only uses the all-right/all-wrong grid matching criterion, ignoring cases where some reasoning is correct or intermediate steps are reasonable, limiting the fine-grained analysis of model capabilities.

**Questions:**

1. see weaknesses

2. In your experiments with supplementary visual information, you found that text-only input outperformed both single-image and multi-image formats. I'm curious: what happens if you combine visual input with Chain of Thought (CoT) cues? Would there be a performance boost?

---

> ### Author Response · Authors · 2025-11-21
> **Response Part 1**
>
> Dear Reviewer K9CV：
>
> Thank you for appreciating our paper as a diverse and explicitly structured work. Your constructive comments and suggestions are valuable to us. Below is our detailed response to clarify the points and answer the concerns you raised.
>
> ### **W1&Q1: Clarification that we have provided a fine-grained analysis of model capabilities.**
>
> Thank you for your helpful suggestion. Actually, we have considered this aspect and conducted a fine-grained analysis of model capabilities in **Appendix E.2**. In addition to the mentioned all-right/all-wrong grid-matching accuracy, we further designe and compute two additional metrics: the grid size precision and grid matching percentage, as reported in **Appendix E.2 Table 5**. Below, we describe the definitions of the two metrics:
>
> **(1) Grid Size Precision:** Referring to the previous study [1], we employ the grid size precision to check if the size of LLM's output grid matches the size of the ground truth grid. If the two sizes match, the grid size precision will be scored as 1; otherwise, it will be scored as 0. This assesses the model's ability to handle grid dimensions.
>
> **(2) Grid Matching Percentage:** This metric represents the proportion of matching elements between the LLM response grid and the ground truth grid. In some cases, the LLM only partially but not fully understands and applies the abstract reasoning rule. Therefore, this grid matching percentage can offer a more fine-grained and more nuanced differentiation of the LLM’s reasoning ability. Notably, if the grid sizes are unequal, the grid matching percentage is directly set to 0.
>
> As **Table A**, most models have very high grid size precision, indicating they can generally infer the correct size of the output grid. The grid-matching percentage is also relatively high, indicating models often produce outputs similar but not fully matched with the ground truth. What’ more, both the grid size precision and grid matching percentage generally decrease as cognitive level increases, consistent with the original accuracy, validating our data framework.
>
> Table A: The average results of grid size precision/grid matching percentage/original accuracy in four levels.
>
> | Model | Size | Count | Shape | Avg-1 | Rotation | Move | Symmetry | Avg-2 | Category | Sort | Planning | Avg-3 | Optics | Mechanics | Thermal | Avg-4 |
> | --- | --- | --- | --- | --- | --- | --- | --- | --- | --- | --- | --- | --- | --- | --- | --- | --- |
> | **General LLMs** |  |  |  |  |  |  |  |  |  |  |  |  |  |  |  |  |
> | Claude-3.7 | 100/99/65 | 100/91/63 | 100/42/13 | 100/83/58 | 100/88/68 | 99/64/57 | **100/89/49** | 99/78/58 | **100/73/54** | 100/94/2 | **100/88/54** | **100/83/44** | **100/61/8** | 100/75/15 | 100/59/0 | 100/65/7 |
> | Qwen3-32B | 91/90/61 | **100/95/71** | 100/45/18 | 96/82/60 | 100/67/51 | 90/42/29 | 36/20/1 | 77/43/27 | 85/57/7 | 83/77/3 | 100/63/8 | 88/64/7 | 64/22/0 | 100/50/0 | 100/51/0 | 88/41/0 |
> | GPT-4o | 100/89/62 | 100/84/44 | 100/40/13 | 100/76/51 | 99/59/27 | 95/10/3 | 86/65/2 | 93/40/9 | 98/66/8 | 100/95/2 | 98/64/8 | 98/73/7 | 96/47/0 | 100/59/0 | 98/40/0 | 98/49/0 |
> | Qwen2.5-32B | 72/61/44 | 100/78/28 | 100/29/6 | 89/60/35 | 67/18/5 | 17/1/0 | 5/3/0 | 28/6/1 | 91/54/4 | 63/58/1 | 93/38/7 | 84/51/4 | 96/42/0 | 93/34/0 | 66/33/0 | 85/36/0 |
> | **Reasoning LLMs** |  |  |  |  |  |  |  |  |  |  |  |  |  |  |  |  |
> | o1 | 99/97/64 | 100/88/60 | **100/65/58** | **99/86/62** | **100/97/93** | **94/76/69** | 64/53/6 | **87/75/58** | 87/71/26 | **100/94/11** | **100/86/53** | 94/81/28 | 96/52/0 | 100/60/7 | 100/62/0 | 98/58/2 |
> | DeepSeek-R1 | 99/99/60 | **100/95/69** | 100/24/8 | 99/80/57 | **100/89/82** | **95/85/78** | **92/81/16** | **92/81/62** | **100/89/44** | 100/90/0 | 100/86/44 | 100/89/35 | 100/57/0 | 100/53/1 | 100/58/0 | 100/56/0 |
> | o1-mini | 85/83/40 | 100/93/65 | 100/43/18 | 94/78/46 | 90/69/63 | 63/36/32 | 17/10/0 | 57/38/31 | 70/56/43 | **76/72/7** | 97/74/43 | **79/65/36** | 76/29/0 | 22/9/0 | 80/47/0 | 59/28/0 |
> | o3-mini | 78/71/31 | 99/92/60 | **100/78/71** | 91/81/45 | 82/56/50 | 55/23/20 | 21/14/1 | 53/30/23 | 54/42/25 | **78/74/7** | 91/47/25 | 71/52/21 | 76/36/0 | **100/73/31** | 73/42/0 | **83/50/10** |
> | QwQ-32B | **94/94/78** | 100/95/61 | 100/35/13 | **97/81/65** | 100/82/64 | 85/34/22 | 88/64/4 | 90/57/29 | 88/62/12 | 92/86/0 | 100/79/34 | 92/73/14 | 100/44/0 | 93/37/0 | 82/31/0 | 91/37/0 |
> | SkyWork-OR1-32B | 93/92/59 | 100/95/68 | 100/43/13 | 97/82/57 | 100/85/64 | 64/27/15 | 94/71/4 | 83/57/25 | 96/62/9 | 100/92/0 | 100/80/36 | 98/75/12 | 100/44/0 | 96/43/0 | 2/0/0 | 66/29/0 |
> |  |

---

> ### Author Response · Authors · 2025-11-21
> **Response Part 2**
>
> ### **Q2: What happens if combining visual input with CoT cues?**
>
> Thank you for your valuable proposal, and it’s our pleasure that you are interested in the experiments about supplementary visual information. The Chain-of-Thought (CoT) is indeed a widely recognized [2,3,4] helpful cue in enhancing the reasoning ability of language models. Following your suggestion, we conduct new experiments to further investigate the impact of visual input with CoT on abstract reasoning tasks.
>
> **(1) Specifically, we conducted further ablations on models, evaluating their performance under the CoT-enhanced prompts setting and the non-CoT prompts setting.**
>
> In the original non-CoT condition, we only employ a basic prompt that asks the model to infer the correct output from the provided training input–output pairs and a test input, together with the associated visual image. Conversely, in the CoT condition, we extended the basic prompt by appending an additional CoT instruction, which required the models to generate detailed reasoning processes. To ensure experimental completeness, we employed two widely used CoT instructions: *“let’s think step by step”* [3] and *“let’s think step by step and reflect on whether your answer is correct”* [4].
>
> (2) **The experimental results under the four settings — with/without visual input and with/without CoT cues — are shown in Table B**.
>
> Compared with the only-text setting, the accuracy of combining visual information with CoT occasionally yields improvements in GPT-4o.  But in general, the differences among the three settings——the text-only, adding single-image/multi-image visualization, and combining CoT prompts——, are not very substantial. In some visual settings, the accuracy is even slightly lower than that of the text-only setting.
>
> We speculate that this may be due to two points: (1) Prior studies [5,6] reveal that when solving multi-modal problems, LVLMs may tend to rely more on textual cues while underutilizing or overlooking visual signals. (2) Some abstract reasoning grids are relatively large, so their visualizations are dense block-like images. Prior studies [5, 7] have revealed that current LVLMs still struggle with grounding and understanding images containing dense information. Therefore, leveraging purely visualization to assist abstract reasoning remains challenging for models. **We have updated Table 2 with CoT results in the revised paper.**
>
> Table B: Comparison of accuracy across four levels of cognitive reasoning, under the text only (only-text), single image(S-Img), multi-image(M-Img) settings, and with CoT1/CoT2 cues settings.
>
> | Model | Vision | L-1 | L-2 | L-3 | L-4 |
> | --- | --- | --- | --- | --- | --- |
> | GPT-4o |  |  |  |  |  |
> |  | only-text | **88.42** | 2.86 | 5.00 | 0.00 |
> |  | S-Img | 78.95 | 1.44 | 0.00 | 0.00 |
> |  | S-Img + CoT1 | 78.87 | 1.17 | 0.00 | 0.00 |
> |  | S-Img + CoT2 | 79.11 | 2.86 | 0.00 | 0.00 |
> |  | M-Img | 74.74 | 8.57 | 5.00 | 0.00 |
> |  | M-Img + CoT1 | 74.82 | **8.82** | 4.50 | 0.00 |
> |  | M-Img + CoT2 | 77.29 | 7.86 | **5.50** | 0.00 |
> | Claude-3.7 |  |  |  |  |  |
> |  | only-text | 95.26 | **25.71** | **45.00** | **15.87** |
> |  | S-Img | 96.84 | 17.14 | 31.25 | 15.87 |
> |  | S-Img + CoT1 | 96.97 | 16.33 | 31.75 | 15.87 |
> |  | S-Img + CoT2 | 95.28 | 18.71 | 33.5 | 16.67 |
> |  | M-Img | **97.89** | 17.14 | 35.00 | 12.70 |
> |  | M-Img + CoT1 | 96.73 | 17.14 | 34.50 | 14.13 |
> |  | M-Img + CoT2 | 97.14 | 16.86 | 35.25 | 13.33 |
> |  |
>
> ## **References**
>
> [1] Wu, J., Yu, M., Liu, L., Yeung, D. Y., & Zhou, J. (2025). Understanding LLMs' Fluid Intelligence Deficiency: An Analysis of the ARC Task. *arXiv preprint arXiv:2502.07190*.
>
> [2] Wei, J., Wang, X., Schuurmans, D., Bosma, M., Xia, F., Chi, E., ... & Zhou, D. (2022). Chain-of-thought prompting elicits reasoning in large language models. *Advances in neural information processing systems*, *35*, 24824-24837.
>
> [3]  Kojima, T., Gu, S. S., Reid, M., Matsuo, Y., & Iwasawa, Y. (2022). Large language models are zero-shot reasoners. *Advances in neural information processing systems*, *35*, 22199-22213.
> [4] Shinn, N., Cassano, F., Labash, B., Gopinath, A., Narasimhan, K., & Yao, S. (2023). Reflexion: Language agents with verbal reinforcement learning, 2023. *URL https://arxiv. org/abs/2303.11366*, *1*.
>
> [5] Sim, M. Y., Zhang, W. E., Dai, X., & Fang, B. (2025, July). Can vlms actually see and read? A survey on modality collapse in vision-language models. In *Findings of the Association for Computational Linguistics: ACL 2025* (pp. 24452-24470).
>
> [6] Deng, A., Cao, T., Chen, Z., & Hooi, B. (2025). Words or Vision: Do Vision-Language Models Have Blind Faith in Text?. In *Proceedings of the Computer Vision and Pattern Recognition Conference* (pp. 3867-3876).
>
> [7] Zhu, H., Su, L., Mao, S., & Ye, J. (2025, January). Read before grounding: Scene knowledge visual grounding via multi-step parsing. In *Proceedings of the 31st International Conference on Computational Linguistics* (pp. 1136-1149).

---

> > ### Comment · Reviewer_K9CV · 2025-11-21
> > **Reply to author's responses**
> >
> > Thank you for the authors' detailed response. It has effectively addressed my questions, and I consider this to be a solid and insightful piece of work.

---

> ### Author Response · Authors · 2025-11-22
> **Thanks for your thorough review**
>
> Dear Reviewer K9CV,
>
> We are deeply grateful for your thorough review and valuable comments, and we are very happy to be recognized as a solid and insightful piece of work! Your suggestions have been instrumental in refining our paper. It is our luck to discuss with such a meticulous and knowledgeable reviewer as you!

---

### Official Review · Reviewer_QK7j · 2025-10-28

**Soundness:** 1
**Presentation:** 3
**Contribution:** 2
**Rating:** 2
**Confidence:** 4

**Summary:**

This paper introduces DRE-Bench, a dynamic reasoning evaluation benchmark designed to assess the fluid intelligence of LLMs. The benchmark consists of 36 abstract reasoning tasks organized across four cognitive levels, with each task featuring multiple dynamic variants. The authors evaluate a range of LLMs, including general-purpose models and reasoning-specialized models. Experimental results reveal that while most LLMs perform well in low-level cognitive tasks, they struggle with high-level cognitive tasks, exhibiting limited generalization capabilities.

**Strengths:**

Proposes an abstract reasoning framework based on cognitive levels.
Designs verifiable code generators and solvers to ensure data quality and scalability.

**Weaknesses:**

Inadequate Human Experiment Design:
Alignment with human cognition is the core advantage of this Benchmark. However, the handling of this critical aspect is very weak in this paper. Although a human comparison experiment is provided, there is a lack of detailed age distribution and significance testing. Additionally, there is no explanation of how participants were motivated to complete the questionnaire seriously or how invalid responses were excluded. For the complex task of designing cognitive test questionnaires, the paper seems to lack rigor; neither the main text nor the supplementary materials provide detailed explanations of how the questionnaire design excludes potential influencing factors. This leads to the conclusion "validates the justification of our 4-level framework" appearing unreliable. This point is particularly important because if the foundation of the human experiment is unreliable, the entire Benchmark's cognitive alignment will be questioned, thereby affecting its validity as a Benchmark.

Lack of Detailed Analysis of Model Failure Modes: Specifically, error types in high-level cognitive tasks are not thoroughly explored, failing to reveal the specific reasons for model failures in complex tasks.

Key Findings Lack Depth: The paper emphasizes five key findings, but these findings do not demonstrate particularly outstanding innovation or in-depth analysis, appearing rather superficial.

Insufficient Analysis of Visual Information Impact: This is one of the key challenges in human and LLM testing. However, the analysis of how visual information affects model performance is overly simplistic, providing results for only two visualization formats and lacking deeper exploration.

**Questions:**

Why were only 20 annotators selected as the sample for the human comparison experiment? This number appears particularly insufficient given the multitude of potential influencing factors in this experiment. Since this is a questionnaire-based test, recruiting more participants would not have been an expensive or labor-intensive task, yet it could have significantly improved the reliability and statistical robustness of the results.

Is there sufficient statistical significance to support the conclusions?
How was the seriousness of participants and the validity of the questionnaire ensured? Were any incentives or quality control measures implemented?
Were potential cognitive biases or other influencing factors considered during the questionnaire design process? How were these interferences excluded to ensure the reliability of the experimental results?
What are the specific failure modes of the models in high-level cognitive tasks? Are there certain types of errors that repeatedly occur?

**Details Of Ethics Concerns:**

The study involved 20 human participants, who should be regarded as human subjects rather than merely treated as annotators. This omission directly contradicts the authors' claim that "The study involves no human subjects." The research failed to address critical ethical considerations, such as providing compensation, outlining detailed experimental procedures, or obtaining informed consent.

---

> ### Author Response · Authors · 2025-11-21
> **Response Part 1**
>
> Dear Reviewer QK7j,
>
> Thank you for recognizing that our verifiable code generators and solvers can ensure data quality and scalability. Your constructive comments and suggestions are valuable to us. Below is our detailed response to address your concerns.
>
> ### **W1&Q1: More detailed design and statistical analysis of Human Experiment.**
>
> Thanks for your valuable comment, and we are glad that you recognize cognitive alignment as a key advantage of our work. To address your concerns, below we provide additional design details and statistical analysis of human experiments. **These analyses have been updated to Appendix E.9 in the revised paper.**
>
> ### **(1) Detailed age distribution of the human experiment**
>
> We further analyze the age distribution of participants in the human study, and the results are presented in **Figure 17 of the revised paper**. We can see that the age distribution of our participants is relatively balanced, embodying a certain degree of representativeness.
>
> ### **(2) Significance testing of the human experiment**
>
> To perform significance testing, we calculate the standard deviation of human accuracy on different tasks and conduct an independent t-test between the two distributions of human and model accuracy.
>
> - **The standard deviation of human accuracy:** The results of **Table A** show the standard deviation among human participants in different tasks. Most standard deviations belong to a normal range (less than 3),  which can validate the credibility of our human study.
>
> Table A: The mean and standard deviation among human participants in different tasks.
>
> |  | level-1 |  |  | level-2 |  |  | level-3 |  |  | level-4 |  |  |
> | --- | --- | --- | --- | --- | --- | --- | --- | --- | --- | --- | --- | --- |
> | Result | Size | Count | Shape | Rotation | Move | Symmetry | Category | Sort | Planning | Optics | Mechanics | Thermal |
> | **Mean** | 75.56 | 82.22 | 68.89 | 91.11 | 75.56 | 46.67 | 73.33 | 24.44 | 88.89 | 46.67 | 77.78 | 17.78 |
> | **Std** | 0.748 | 2.002 | 2.119 | 1.446 | 1.428 | 2.145 | 1.118 | 1.446 | 1.483 | 2.650 | 2.910 | 3.035 |
> |  |
>
> - **Independent t-test between the two distributions of human and model accuracy**: Furthermore, based on the evaluation results of models and humans, we conducted an independent t-test on their distributions. The p-values for all tasks are reported in **Table B**, where the majority of values are below 0.05. These results demonstrate the statistical significance of humans’ and models’ results on our proposed benchmark, validating that the claimed conclusion—humans relatively outperform current models—is well supported.
>
> Table B: Independent t-test between the two distributions of human and model accuracy (t-statistic / p-value).
>
> |  | level-1 |  |  | level-2 |  |  | level-3 |  |  | level-4 |  |  |
> | --- | --- | --- | --- | --- | --- | --- | --- | --- | --- | --- | --- | --- |
> | Result | Size | Count | Shape | Rotation | Move | Symmetry | Category | Sort | Planning | Optics | Mechanics | Thermal |
> | **t-statistic** | -2.8326 | -1.2005 | -3.1882 | -1.5778 | -3.5937 | -3.4689 | -1.6457 | -2.3095 | -4.7075 | -3.8381 | -8.4959 | -3.4058 |
> | **p-value** | 0.0103 | 0.2439 | 0.0046 | 0.1303 | 0.0018 | 0.0024 | 0.1155 | 0.0317 | 0.0001 | 0.001026 | 4.5463e-8 | 0.0028 |
> | **p-value<0.05** | √ |  | √ |  | √ | √ |  | √ | √ | √ | √ | √ |
> |  |
>
> ### **(3) How participants were motivated to complete the questionnaire seriously**
>
>  To motivate participants to complete the questionnaire, we not only offered salaries to participants but also provided detailed and rigorous instructions to them, as described below:
>
> - **Salary:** We have provided a salary of 30 $ per hour to each participant who completed the questionnaire, encouraging them to carefully solve the abstract problems.
> - **Detailed instructions:** We provided very clear instructions in the questionnaire README, which explained the definition of abstract reasoning tasks, identified the given training samples, and explained how to fill in the answers for the test samples. Such explicit and detailed instructions improve the usability of the questionnaire and ensure the rigor of the human study. The specific instruction is as follows:
>
> ```jsx
> Your task is to complete a questionnaire consisting of 400 questions. Each question is an abstract reasoning task. To solve each question, you can follow the steps below:
> 1, Reason out the transformation rule by analyzing the three training sample pairs, namely how the input grids are converted into corresponding output grids.
> 2, Apply the transformation rule to the test input grid and generate the correct test output grid.
> For convenience, the test output grid has been pre-initialized with the content of the test input grid, so you can directly perform the rule-based transformation.
> When you finish a question, click "Next"  to the next one. After completing all questions, click "Final Submit."
> ```

---

> ### Author Response · Authors · 2025-11-21
> **Response Part 2**
>
> **After completion, we further conduct a quality check.** Specifically, we randomly sample 10 participants and compute their average accuracy on the tasks. If the accuracy is substantially lower than the overall average of the selected 10 participants, the participant is flagged as completing without sufficient effort. Fortunately, our participants showed strong compliance, with no abnormally low scores observed, so their results can be regarded as valid samples.
>
> ### **(4) How is the questionnaire designed to exclude potential influencing factors**
>
> We carefully designed the UI interface for the questionnaire and have open-sourced the UI code at https://anonymous.4open.science/r/DRE-Bench-8098. To exclude potential influencing factors in the human study, we adopted several user-friendly measures in the questionnaire design, mainly as follows:
>
> - **General UI interface:** As the UI example in **Figure 18 in the revised paper**, the first three rows correspond to the training sample pairs, where the input grid is shown on the left and the output grid on the right. In the fourth row, the left side displays the test input grid, while the right side is the test output grid that participants must fill in.
>
> - **To mitigate visual overload caused by large grids:** Many of the abstract reasoning tasks involve relatively large grids, often exceeding 15×15 in size. As previous research [1], humans could easily lose track of numerical positions within the grid, causing errors even when they correctly understood the rule. In contrast, LLMs are less affected due to their strong ability to restate longer contexts[2,3]. To fill this gap, we added row and column numbers to the test input grid in the UI, making it easier for humans to locate specific positions (as shown in the red box in Figure 19).
>
> - **To reduce the unnecessary burden of reproducing large grids:** In abstract reasoning tasks, the ground truth output is derived from applying a transformation rule on the test input. Therefore, in many tasks, a large portion of the output grid remains identical to the input. Since the restate ability of models is stronger than humans [2,3], we initialized the output grid with the input grid, helping humans to focus primarily on the reasoning logic and rule transformation. (as shown in the red box in Figure 20).
>
> - **To highlight the actual reasoning operations:** To further emphasize the reasoning logic, we visually highlighted all cells in the output grid that differ from the input grid in bright blue. This provides a user-friendly and intuitive way to visualize the transformation human induced. (as shown in the red box in Figure 21).
>
> In summary, the final questionnaire version has undergone multiple rounds of review by the co-authors. To exclude potential confounding factors, we have carefully designed UI enhancements to make the questionnaire more accessible to human participants.

---

> ### Author Response · Authors · 2025-11-21
> **Response Part 3**
>
> ### **W2&Q2: Lack of Detailed Analysis of Model Failure Modes.**
>
> Thank you for your interest in the error patterns. Following your suggestion, we conducted a deeper investigation into the common error patterns in the high-level tasks (Level-3 and Level-4).
>
> **In detail,** we first visualized the model outputs for the Level-3 and Level-4 tasks and found that the error patterns are highly diverse. Next, we compare errors made by different models on the same task, and summarize some common error patterns from the *Category*, *Planning*, *Gravity*, and *Reflection* tasks, as illustrated in **Figure 16 of the revised paper**:
>
> **(1) Category Task:** In this task, most models were able to correctly identify the object blocks, but failed to fill in the correct colors. A common issue was misunderstanding the color-to-category correspondence (error 1, 2, 3). Another issue was missing objects during the recoloring process, where some objects were left uncolored (error 4). However, most models were able to complete part of the coloring task, indicating that they correctly identified some of the object categories.
>
> **(2) Planning Task:** Here, many models struggled with the required path-planning ability. A large portion of cases produced redundant connections, causing the wrong loops instead of a directed path (error 1, 2, 3). Conversely, others produced incomplete connections, only managed to construct partial paths(error 4). This may be because the planning task involves multiple steps, which creates substantial difficulty for models.
>
> **(3) Gravity Task:** In this task, several cases captured the basic physical sense that objects should move downward, but how objects interact with the ground was captured incompletely. Some cases caused the object to inadequate-descend (error 1, 2, stopping in mid-air), whereas others caused it to over-descend (error 3, 4, passing through the ground), showing that they failed to correctly infer the concept of gravity.
>
> **(4) Reflection Task:** Most cases did not get the rule that a light ray should reflect upon hitting obstacles. Some outputs simply restated the input pattern (error 1, 2), while others produced symmetric patterns that is irrelevant to the intended reflection (error 3, 4). Generally, models perform poorly on this task, and compared with gravity, they show an even weaker understanding of the physical concept of reflection.
>
> ### **W3: Key Findings Lack Depth.**
>
> Thank you for expecting our work to a high standard. In fact, most of our key findings are novel topics that have not been explored in prior research [4,5,6]. **Reviewer K9CV** described our work as *“novel,”* and **Reviewer mbfn** regarded our finding on directions to be *“interesting.”* Moreover, our contributions include many aspects and are not limited to the key findings alone.
>
> **(1) Firstly**, **the listed key findings are derived from the newly proposed cognitive–psychological framework, making them inherently embodied novelty.**
>
> Prior abstract reasoning research [4,5,6] has not aligned different levels with humans. Therefore, the perspectives of different cognitive dimensions, task complexity on such tasks are newly explored by us. Moreover, we also dig into how the number of different in-context training samples, and the inference-time scaling impact on model performance on abstract reasoning tasks.
>
> **(2) Secondly, our contributions include many aspects.**
>
> As **lines 126-133**, in addition to the key findings, we also include many other contributions: we have introduced a human-aligned cognitive framework for abstract reasoning tasks, and developed a verifiable data engine that can scale abstract reasoning data with different complexities.
>
> **(3) Finally**, **beyond the key findings summarized in the Introduction, extra interesting observations are revealed in the experiments section**.
>
> - In **Section 4.5** and **Table 3**, we have explored the Spatial Orientations and surprisingly found that current models may demonstrate a distinct understanding of spatial orientation compared to humans. Models achieve higher and more consistent accuracy in vertical (up/down) directions than in horizontal (left/right) ones in Move.
> - What’s more, in the symmetry task of **Table 3**, models generally performed better for horizontal symmetry than for vertical symmetry. However, from the perspective of human cognition [7,8], directional distinctions are typically perceived as equivalent to humans. However, it seems that models exhibit varying performance across different dimensions of abstract reasoning.
> - In addition, we further exhibit the different error patterns output by models, as detailed in the **response W2**.

---

> ### Author Response · Authors · 2025-11-21
> **Response Part 4**
>
> ### **W4: Insufficient Analysis of Visual Information Impact.**
>
> Regarding the visualization formats, we would like to clarify the following points:
>
> **(1) Firstly**, **we should clarify that visual information may not be the key challenge in human or LLM testing.** Except for the experiment on the Impact of Auxiliary Visual Information, all other human and model testings are conducted using the pure text grid format, which is the official format of such abstract reasoning tasks [4]. The grid-style visualization in our paper (like Figures 1, 2, 3, etc.) is intended solely to help readers better understand the underlying reasoning rules, as in prior works also visualized[4,5,6].
>
> **(2) Secondly,** **our visualization format is the official visualization format used in abstract reasoning works [4,5,6],** which utilizes different colors to represent each grid number and displays the entire grid as a grid image. Therefore, our visualization design is not overly simplified—it adheres to the most appropriate and commonly followed standard within this domain.
>
> **(3) Thirdly,** we agree with your point that richer visual formats are important, so we have carefully designed the *single-image* and *multi-image* settings. Actually, these two visualization formats already contain differences in the granularity of images, the number of images, and the grid-image correspondence. Beyond these, we have not identified additional meaningful visualization approaches. If you have specific alternative formats, we would be very willing to conduct experiments immediately.
>
> ### **References**
>
> [1] Baddeley, A. (2012). Working memory: Theories, models, and controversies. *Annual review of psychology*, *63*(1), 1-29.
> [2] Hsieh, C. P., Sun, S., Kriman, S., Acharya, S., Rekesh, D., Jia, F., ... & Ginsburg, B. (2024). RULER: What's the Real Context Size of Your Long-Context Language Models?. *arXiv preprint arXiv:2404.06654*.
>
> [3] Bai, Y., Lv, X., Zhang, J., Lyu, H., Tang, J., Huang, Z., ... & Li, J. (2024, August). Longbench: A bilingual, multitask benchmark for long context understanding. In *Proceedings of the 62nd Annual Meeting of the Association for Computational Linguistics (Volume 1: Long Papers)* (pp. 3119-3137).
>
> [4] Chollet, F. (2019). On the measure of intelligence. *arXiv preprint arXiv:1911.01547*.
>
> [5] Wu, J., Yu, M., Liu, L., Yeung, D. Y., & Zhou, J. (2025). Understanding LLMs' Fluid Intelligence Deficiency: An Analysis of the ARC Task. *arXiv preprint arXiv:2502.07190*
>
> [6] Yu, M., Liu, L., Wu, J., Chung, T. T., Zhang, S., Li, J., ... & Zhou, J. (2025). The Stochastic Parrot on LLM's Shoulder: A Summative Assessment of Physical Concept Understanding. *arXiv preprint arXiv:2502.08946*.
>
> [7] Aflalo, T. N., & Graziano, M. S. (2008). Four-dimensional spatial reasoning in humans. *Journal of Experimental Psychology: Human Perception and Performance*, *34*(5), 1066.
> [8] Ambinder, M. S., Wang, R. F., Crowell, J. A., Francis, G. K., & Brinkmann, P. (2009). Human four-dimensional spatial intuition in virtual reality. *Psychonomic bulletin & review*, *16*(5), 818-823.

---

> > ### Comment · Reviewer_QK7j · 2025-11-23
> >
> > Thank you for taking the time to write a nice response. This reply has addressed several questions, but there are still some core issues that remain unresolved:
> >
> > I have reviewed the code repository https://anonymous.4open.science/r/DRE-Bench-8098 and it seems that I couldn't find the UI code. It could be due to my oversight, if possible, please provide further clarification on its location. (The readability of the code is relatively poor, and there are even three readme files with identical content. However, this is not a point of deduction.)
> >
> > Regarding the quality check "Specifically, we randomly sample 10 participants and compute their average accuracy on the tasks. If the accuracy is substantially lower than the overall average of the selected 10 participants, the participant is flagged as completing without sufficient effort." This does not seem to be the most common practice for controlling questionnaire quality. I am not sure whether this approach is sufficient for this task. If possible, I hope the authors can add references and citations regarding this practice. Also, I am uncertain about how "substantially lower" is defined? And why choose 10 participants (50%)?
> >
> > I still believe that 20 samples are seriously insufficient for this experiment, and the consideration of experimental samples appears rather rough, especially considering that different age groups may have different cognition due to occupation, major, etc.
> >
> > I am not sure what "visual information may not be the key challenge in human or LLM testing" specifically means. For humans, if visual information is not necessary, then adding visual elements seems to be testing for the sake of testing?
> >
> > I noticed the further analysis of error patterns, but some analyses of error models do not seem convincing enough. For example, in the Gravity Task, is inadequate-descend or over-descend definitely due to failing to correctly infer the concept of gravity? Further analysis or task decomposition seems to be needed here. This will greatly enhance the quality of the paper, but I agree that this seems to be an excessive workload for a paper. If this issue remains unresolved, it is acceptable.
> >
> > This article should be a high-quality work, provided that there is sufficient evidence to prove that this test is consistent with and effective for humans.

---

> ### Author Response · Authors · 2025-11-24
> **Response Part 5**
>
> Thank you for your thorough reading of our work. It is truly our privilege to engage in a discussion with such a meticulous and knowledgeable reviewer as you! Below, we will further conduct experiments and provide additional explanations to address your concerns.
>
> ### **Concern-1: Regarding the code repository.**
>
> Apologies for the inconvenience caused. There was a slight delay in the anonymous repository link, resulting in a synchronization issue. We have now updated the anonymous repository and can confirm that the specific UI code is available at https://anonymous.4open.science/r/DRE-Bench-8098 . What’s more, regarding the readability, we have rewritten the GitHub repository's README file to provide clearer introductions. Thank you for your thoughtful suggestions. We believe these improvements will make our dataset and code more convincing.
>
> ### **Concern-2: Regarding the quality check.**
>
> Our quality check is designed to prevent low-quality questionnaires from affecting the final results. This approach is inspired by previous work [1,2,3] in processing human study. Some studies [1, 2] removed low-engagement participants by applying fixed accuracy thresholds (e.g., 60%, 50%), and some work [3] treated ‘failure to complete all tasks’ as a removal criterion. In our experiment, if a participant left a case unfinished, the response would simply not match the ground truth and thus be scored as 0, and be included in the overall score. Therefore, following cognitive psychology and crowdsourcing research [4,5,6], we defined a relative threshold as those whose accuracy fell more than two standard deviations below the group mean(mean – 2SD).
>
> Based on your feedback, we extended the quality check to all participants. Fortunately, their scores are also higher than the relative threshold (mean – 2SD), which may be attributed to the fact that we compensated participants for their time, encouraging them to complete the task seriously.
>
> ### **Concern-3: Regarding the number of human participants**.
>
> Thank you for your valuable feedback regarding the human study. In fact, after reviewing your comments in the first rebuttal turn, we recognized the importance of the sample size in human experiments. Therefore, we have started to scale to 40 participants, but we have not yet finalized the results due to the time cost of human experiments. **Once the additional human experiments are all completed and the data is processed, we will update the statistical data and results for you.**

---

> > ### Author Response · Authors · 2025-11-24
> > **Response Part 6**
> >
> > ### **Concern-4: Regarding the impact of visual information.**
> >
> > What we meant by 'Firstly, we should clarify that visual information may not be the key challenge in human or LLM testing' is a response to your initial comment, 'Insufficient Analysis of Visual Information Impact: This is one of the key challenges in human and LLM testing.' If this initial comment has some other meanings, please let us know, and we will further provide a more detailed explanation for you.
> >
> > As we have clarified, the official format for abstract reasoning tasks is purely text-based. This task was initially designed as text-only, so visual information is not necessary for either models or humans. However, in our work, exploring visual information is not to be testing for the sake of testing; on the contrary, it holds significant value. The meanings for exploring visual information are as mainly follows:
> >
> > 1) **Although the abstract reasoning task is initially a text-only question, it is often discussed alongside IQ tests and classic visual puzzles [7,8,9].** Such abstract reasoning task and well-known visual puzzles (like Raven’s Progressive Matrices[8] and Bongard problems[9]) are regarded as the same kind of human-intelligence-oriented inductive reasoning problems, whose purpose is to evaluate few-shot concept abstraction and pattern induction abilities[10].  The authoritative surveys on abstract visual reasoning [11] even place ARC in a separate category, referring to it as a 'new generation of abstract visual reasoning benchmarks'. Therefore, we investigate the impact of auxiliary visual information on abstract reasoning is highly meaningful and reasonable.
> >
> > 2) **A series of recent studies [12,13,14] have also started to explore visual approaches for abstract reasoning tasks.** The recent work by Kaiming He’s group [12] treats ARC as a visual problem by mapping the ARC grid onto a 'canvas' and performing image-to-image translation as in standard vision tasks. The ConceptARC Benchmark [13] also evaluates ARC tasks on GPT-4V by directly feeding images, but found the accuracy remained very low. In addition, survey [14] argues that solving ARC should incorporate visual imagery, mimicking the human strategy of mentally simulating transformations. These works have revealed that the visual component in ARC is increasingly drawing attention, and the community is not necessarily limited to a text-only problem-solving approach. Therefore, our experiment to explore whether auxiliary visual information can help abstract reasoning is very promising.
> >
> > ### **Concern-5: Regarding the error patterns of gravity.**
> >
> > Thank you for your interest in error patterns. Regarding the issue of insufficient descent or over-descent in the gravity task, prior work[15] has discussed relevant phenomena of objectness priors, which emphasize handling objects and their interactions. Physical contact between objects is a recurring theme (e.g., gravity-related transformations or objects expanding until they collide with others). Therefore, we think one possible explanation is that the model did not correctly infer these physical interactions, but other factors may also contribute.
> >
> > Moving forward, as you suggested, we will continue to investigate how to improve accuracy on ARC tasks and reduce such error patterns.

---

> > > ### Author Response · Authors · 2025-11-24
> > > **Response Part 7**
> > >
> > > ### **References**
> > >
> > > [1] Lawrence, R. K., Cochrane, B. A., Eidels, A., Howard, Z., Lui, L., & Pratt, J. (2023). Emphasizing responder speed or accuracy modulates but does not abolish the distractor-induced quitting effect in visual search. *Cognitive research: principles and implications*, *8*(1), 63.
> > > [2] Garre-Frutos, F., Vadillo, M. A., González, F., & Lupiáñez, J. (2024). On the reliability of value-modulated attentional capture: An online replication and multiverse analysis. *Behavior Research Methods*, *56*(6), 5986-6003.
> > >
> > > [3] Kurvers, R. H., Herzog, S. M., Hertwig, R., Krause, J., Moussaid, M., Argenziano, G., ... & Wolf, M. (2019). How to detect high-performing individuals and groups: Decision similarity predicts accuracy. *Science advances*, *5*(11), eaaw9011.
> > >
> > > [4] Schurgin, M. W., Wixted, J. T., & Brady, T. F. (2020). Psychophysical scaling reveals a unified theory of visual memory strength. *Nature human behaviour*, *4*(11), 1156-1172.
> > >
> > > [5] Unsworth, S., & Blom, E. (2010). Experimental methods in language acquisition research.
> > >
> > > [6] Boland, J. E., Atkinson, E., De Los Santos, G., & Queen, R. (2023). What do we learn when we adapt to reading regional constructions?. *Plos One*, *18*(4), e0282850.
> > >
> > > [7] Wechsler, D. (1955). Manual for the Wechsler Adult Intelligence Scale (WAIS) Psychological Corporation. *New York*.
> > > [8] Burke, H. R. (1958). Raven's Progressive Matrices: A review and critical evaluation. *The Journal of genetic psychology*, *93*(2), 199-228.
> > >
> > > [9] Theodoridis, S., & Koutroumbas, K. (2006). *Pattern recognition*. Elsevier.
> > >
> > > [10] Banerjee, S. (2025). Enhancing AI Capabilities on the Abstraction and Reasoning Corpus: A Path Toward Broad Generalization in Intelligence.
> > >
> > > [11] Małkiński, M., & Mańdziuk, J. (2023). A review of emerging research directions in abstract visual reasoning. *Information Fusion*, *91*, 713-736.
> > > [12] Hu, K., Cy, A., Qiu, L., Ding, X. D., Wang, R., Zhu, Y. E., ... & He, K. (2025). ARC Is a Vision Problem!. *arXiv preprint arXiv:2511.14761*.
> > >
> > > [13] Moskvichev, A., Odouard, V. V., & Mitchell, M. (2023). The conceptarc benchmark: Evaluating understanding and generalization in the arc domain. *arXiv preprint arXiv:2305.07141*.
> > >
> > > [14] Ainooson, J., Sanyal, D., Michelson, J. P., Yang, Y., & Kunda, M. (2023). A neurodiversity-inspired solver for the abstraction\& reasoning corpus (arc) using visual imagery and program synthesis. *arXiv preprint arXiv:2302.09425*.
> > >
> > > [15] Bober-Irizar, M., & Banerjee, S. (2024). Neural networks for abstraction and reasoning. *Scientific Reports*, *14*(1), 27823.

---

> ### Author Response · Authors · 2025-11-27
> **Response Part 8**
>
> Dear Reviewer QK7j,
>
> Thank you for your valuable feedback regarding the human study. In fact, after reviewing your comments in the first rebuttal turn, we recognized the importance of the sample size in human experiments. Therefore, we have scaled the human study to 40 participants these days, and now we update a series of experiment results for you.
>
> ### **Updating-1: Regarding the detailed age distribution**.
>
> We further analyze the age distribution of participants in the human study, and the results are updated in **Figure 17 of the newly revised paper**. We can see that the age distribution of our final participants is also relatively balanced, embodying a certain degree of representativeness.
>
> ### **Updating-2: Regarding the standard deviation of human accuracy.**
>
> With the expanded human study (N = 40), the results in **Table C** show the standard deviations across participants for each abstract reasoning task. As shown in **Table C**, the standard deviations of each task remain within a normal range (below 2), and are even more stable compared to the results from the previous participant size (N = 20). Your suggestion has been extremely helpful, and these updated statistics further support the credibility and reliability of our human study.
>
> Table C: The updated mean and standard deviation among human participants in different tasks.
>
> |  | level-1|  |  | level-2 |  |  | level-3 |  |  | level-4 |  |  |
> | --- | --- | --- | --- | --- | --- | --- | --- | --- | --- | --- | --- | --- |
> | Result | Size | Count | Shape | Rotation | Move | Symmetry | Category | Sort | Planning | Optics | Mechanics | Thermal |
> | **Mean** | 75.96 | 82.02 | 71.72 | 84.65 | 77.78 | 44.25 | 75.75 | 29.49 | 89.9 | 49.7 | 76.16 | 16.16 |
> | **Std** | 0.93 | 1.42 | 1.61 | 1.67 | 1.64 | 1.34 | 1.70 | 1.57 | 1.52 | 1.52 | 1.61 | 1.23 |
> |  |
>
> ### **Updating-3: Regarding the Independent t-test between human and model accuracy.**
>
> With the expanded scale of human study (N = 40), we re-conduct the independent t-testing between human and model performance distributions. As reported in **Table D**, the majority of task p-values remain below 0.05, showing that the statistical significance holds under a larger and more convincing experiment size. This updated t-testing further strengthens and supports our conclusion that, while human performance is generally aligned with model behavior across cognitive levels, humans still exhibit a degree of performance advantage over current models on DRE-Bench.
>
> Table D: Updated independent t-testing between the two distributions of human and model accuracy (t-statistic / p-value).
>
> |  | level-1 |  |  | level-2 |  |  | level-3 |  |  | level-4 |  |  |
> | --- | --- | --- | --- | --- | --- | --- | --- | --- | --- | --- | --- | --- |
> | Result | Size | Count | Shape | Rotation | Move | Symmetry | Category | Sort | Planning | Optics | Mechanics | Thermal |
> | **t-statistic** | -3.7147 | -1.1487 | -3.0798 | -1.7772 | -3.8474 | -2.9810 | -2.2172 | -2.2668 | -5.0581 | -3.6903 | -9.9144 | -2.8960 |
> | **p-value** | 0.0014 | 0.2649 | 0.0061 | 0.0915 | 0.0010 | 0.0076 | 0.0390 | 0.0352 | 6.9822e-05 | 0.0015 | 6.0433e-09 | 0.0092 |
> | **p-value<0.05** | √ |  | √ |  | √ | √ | √ | √ | √ | √ | √ | √ |
> |  |
>
> ### **In summary**
>
> **Thanks very much for your continual response and suggestion！ We believe that our discussion with you has been highly valuable and has substantially improved the quality of our work,** especially in further strengthening the credibility and reliability of our human study. **The results of human study have also been updated in the revised paper.** We hope that our responses can address and alleviate your ethics concerns. We are truly pleased to have the opportunity to discuss with you and to improve our paper accordingly.

---

> > ### Comment · Reviewer_QK7j · 2025-11-27
> >
> > Thank you for your detailed responses, which have effectively addressed my concerns. I will update my score to 4. I believe this is a novel piece of work, but I would like to explain why I am not inclined to give a higher score.
> >
> > First, the fact that 100% of the samples are reported as valid raises some questions. This might be due to issues with the quality control method or the inherent discriminative power of the questions themselves.
> >
> > Also, in the references cited by the authors, it can be observed that other quality control methods beyond those used in this paper are typically employed.
> >
> > Second, even with 40 samples, this number remains far from sufficient (and is lower than the sample sizes used in the papers referenced in our discussion). While in AI research, dozens of samples may sometimes be acceptable, the core argument of this paper is that previous works have failed to align well with human factors.
> >
> > Additionally, this alignment should consider multiple factors, whereas this paper focuses solely on age.
> >
> > Lastly, regarding the code update (which does not affect the scoring), I noticed several non-English sections (including comments and HTML parts). The reason for this is unclear, but I suggest providing a bilingual version if possible.
> >
> > Thank you again for your thoughtful responses and for addressing my concerns.

---

### Official Review · Reviewer_mbfn · 2025-10-30

**Soundness:** 1
**Presentation:** 2
**Contribution:** 3
**Rating:** 2
**Confidence:** 5

**Summary:**

This paper introduces a large-scale abstract reasoning benchmark structured along a four-level cognitive hierarchy (Attribute, Spatial, Sequential, and Conceptual). The framework is designed to evaluate Large Language Models (LLMs) on tasks of increasing cognitive complexity and to assess whether current models exhibit good fluid intelligence. Through systematically generated tasks and extensive experiments, the authors find that model accuracy declines with higher cognitive levels, especially for physics-related conceptual reasoning. Reasoning-optimized models (e.g., OpenAI-o1, DeepSeek-R1) outperform general-purpose LLMs, and the performance gap widens as task complexity increases.

**Strengths:**

- It is an interesting finding that the models achieve higher and more consistent accuracy in vertical (up/down) directions than in horizontal (left/right) ones in Move. Similarly, in symmetry tasks, performance is better for horizontal symmetry than for vertical symmetry
-  The authors implement a verifiable, scalable data engine capable of generating diverse reasoning tasks with controllable complexity—an important methodological contribution that ensures reproducibility and adaptability.
- The results highlight key cognitive distinctions: (a) model accuracy degrades sharply at higher cognitive levels, (b) inference-time scaling contributes mainly to lower-level reasoning, and (c) visual input may not aid abstract reasoning. These insights deepen our understanding of the limits of current LLM cognition.

**Weaknesses:**

- Several aspects of the experimental design and presentation require clarification and stronger consistency.
 the selection of models across figures lacks transparency and methodological coherence. For example, Figure 5 does not specify why those particular four models were chosen, Figure 6 focuses solely on DeepSeek-R1 without justification, and Figure 7 switches to yet another subset of models while testing only two tasks.

Such inconsistent model selection makes it difficult to assess whether observed relationships are generalizable. A more systematic comparison across models, task types, and difficulty levels would provide stronger evidence for the claimed trends. Table 2 also introduces new models without explanation

- question design and ground-truth validity raise concerns. In Figure 8, certain problems might have multiple correct solutions (e.g., the Level-3 task where the path could start from one red square to the upper blue square not only the bottom one), yet only one is labeled as correct. Additionally, the prompt mislabels the target as a “red dot” instead of a “red square.”

 In the Level-4 example, two obstacle types exist, but the prompt fails to specify that only the blue obstacle is reflective, potentially confusing both human and model respondents. These ambiguities undermine the reliability of the evaluation.

- discrepancies between text and data should be addressed.
- Line 340 claims that human accuracy decreases with task level, but Table 1 shows relatively stable human performance across Levels 1–3, except for a few specific tasks (e.g., sorting). Interestingly, these same tasks are also where models perform poorly. This suggests that further validation is needed to ensure the generated items reflect increasing cognitive difficulty rather than artifacts of task design.

**Questions:**

- What criteria guided the selection of models in Figures 5–7 and Table 2? Were these choices based on availability, performance tiers, or specific architectural features?
- Why was DeepSeek-R1 uniquely highlighted in Figure 6?
- How many tasks and difficulty levels were used to derive the inference-time–accuracy relationship, and do these results hold across other models or scales?
- In Figure 8, how were ground truths defined for multi-path problems, and were alternative valid solutions considered during evaluation?
- How were ambiguous or mislabeled prompts (e.g., “red dot” vs. “red square,” unspecified reflective obstacles) handled in scoring and analysis?
- Given that human performance appears stable across the first three levels, how to justify the statement that accuracy “generally decreases” with increasing cognitive level?

Suggestions:

- Unify model selection and reporting. clearly specify which models are used in each experiment and maintain consistency across figures. If subsets differ, explain the rationale.

- Broaden the inference-time analysis. test multiple models across several tasks and difficulty levels to confirm that observed correlations are not model-specific.

- Validate and disambiguate tasks. review prompts and ground truths to ensure each question has a unique, well-defined solution and that instructions are precise.

- Re-evaluate human baselines. conduct additional validation to verify that increasing task complexity indeed reflects higher cognitive difficulty, rather than inconsistencies in dataset generation.

- Clarify inconsistencies between text and data.

---

> ### Author Response · Authors · 2025-11-21
> **Response Part 1**
>
> Dear Reviewer mbfn,
>
> Thank you for appreciating our paper as a verifiable and scalable data engine, with interesting findings. Your constructive comments and suggestions are valuable to us. Below is our detailed response to address your concerns.
>
> ### **W1&Q1&Q2&Q3: What criteria guided the selection of models in Figures 5–7 and Table 2?**
>
> Thank you very much for your careful reading. We truly appreciate having a reviewer as responsible as you, which gives us the opportunity to explain the criteria that guided the selection of models in experiments. Inspired by your suggestion, we have further conducted more comprehensive experiments to improve our paper. **All the mentioned figures and tables have been added to the revised paper.** Below, we provide a detailed explanation of model selection in Figures 5–7 and Table 2.
>
> ### **(1) The reason and refinement for Figure 5**
>
> **Reason for model selection**: Figure 5 is a scatter plot of model accuracy versus variance, which involves multiple colors representing different models and multiple marker shapes representing different tasks. We initially intended to plot all models in this figure, but we found that doing so resulted in excessive visual clutter. Therefore, in the final Figure 5, we select several top-performing models from **Table 1** to clearly and intuitively present their accuracy versus variance—namely, o1, Claude 3.7, DeepSeek-R1, and QwQ-32B. Meanwhile, we also recognize the importance of showing results for all models. **Therefore, to ensure completeness, we have already included the average performance of all models across four cognition levels in Figure 1(c)**, serving as the Leaderboard of LLM intelligence on our DRE-Bench.
>
> **Refined experiment**: We now additionally present the full results of all models in **Figure 14**, and we are still able to draw similar conclusions to the original paper. As shown in **Figure 14**, for the majority of *Level-1 attribute* tasks, most models demonstrate strong performance and high stability. However, when the task level increases to *Level-2 spatial*, most models exhibit substantial degradation in both accuracy and stability, indicating limited generalization capabilities at this level. In contrast, OpenAI-o1 and DeepSeek-R1 maintain comparable performance to that observed at *Level-1*, highlighting the advantage of reasoning models in solving more cognitively demanding tasks. Moreover, in *Level-3 sequential*, most of the scatter points are concentrated in the lower-left region, suggesting that current models struggle to generalize effectively across the more complex and varied tasks at higher cognitive levels.
>
>  ### **(2) The reason and refinement for Figure 6 and Figure 7**
>
> **Reason for model selection**: We are very grateful for your valuable suggestion. We agree that showing results of more models can indeed provide a deeper exploration of the impact factors. Previously, we randomly selected some top-performing models from **Table 1** as representative models, utilizing DeepSeek-R1 for Figure 6, and o1 for Figure 7. Now, we supplement additional model results for the two ablation experiments: *Impact of the Number of In-context Learning Samples* experiment (Figure 6) and *Impact of the Inference Time* experiment (Figure 7).
>
> **Refined experiment of Figure 6:** We illustrate more sufficient results on all evaluated models and **update Figure 6 in the revised paper**.  To deeply explore how the number of in-context samples affects performance in the abstract reasoning scenario, we conduct experiments on all evaluated models and compute the average accuracy across all models. The average results in **Figure 6** clearly reveal the following two observations: (1) Overall, increasing the number of in-context samples helps models better capture underlying rules and improve performance. (2) In mediumly higher levels like *Level-2 Spatial*, *Level-3 Sequential,* increasing the number of in-context training samples leads to noticeable performance improvements. (3) However, for *Level-1 Attribute* and  *Level-4 Conceptual* tasks, increasing the number of samples yields limited improvement. This suggests that adding more in-context examples has a limited impact when the model has already mastered the easy tasks or lacks the inherent capability to solve too difficult tasks. In summary,  this refined experiment more rigorously validates the conclusions of our original manuscript.
>
> **Refined experiment of Figure 7**: To refine Figure 7, we not only evaluate inference time on more models but also on more tasks across four levels. To obviously reveal the trend of accuracy changing, we select the task with the highest average model accuracy for each level—namely, count, rotation, agent, and mechanics. In addition, the Level-4 mechanics task is so challenging that only Claude 3.7, O1, and O3-mini achieve non-near-zero accuracy. Thus, we expand the inference-time experiments on the four tasks using these three models.

---

> ### Author Response · Authors · 2025-11-21
> **Response Part 2**
>
> As **Figure 15 of the revised paper** shows, in the *level-1 count* task, as task complexity increases, the model tends to engage in deeper reasoning and can effectively maintain relatively stable and high accuracy. In the *Level-2 rotation* task, the models exhibit an overall trend: rotations by single multiples of 45° yield substantially lower accuracy than rotations by multiples of 90°. This is reasonable, as prior studies [1] have demonstrated that it is more difficult for humans to perceive lines at 45° diagonal orientations. However, in high-level tasks (i.e., *level-3,level-4*), even though the model’s inference time increases, it still fails to solve the more complex cases. This indicates that simply increasing inference time is insufficient to compensate for the model’s inherent limitations in high-level reasoning.
>
> ### **(3) The reason and refinement for Table 2**
>
> **Reason for model selection**: As for Table 2, we apologize for not explaining the rationale behind using GPT-4o and Claude-3.7 in the previous version. We now provide a detailed clarification of why selecting these two models for the Auxiliary Visual Information ablation, which is primarily based on the following two conditions. **Firstly,** this ablation study requires models with multimodal capabilities. **Secondly**, the multi-modal setting necessitates that the model support at least seven input images. Among all evaluated models at that time, only GPT-4o and Claude-3.7 simultaneously satisfied the two conditions. Therefore, in Table 2, we utilized these two qualified models to explore the impact of Auxiliary Visual Information.
>
> ### **W2&W3&Q4&Q5: Clarification on question design and ground-truth validity raises concerns.**
>
> Thank you for your careful reading. We would like to clarify that all of our task designs guarantee a unique correct solution, including the Level-3 and Level-4 tasks shown in Figure 8.
>
> ### **(1) Clarifications on ambiguous or mislabeled prompts**
>
> **First, we would like to clarify that the “rules” shown in Figure 8 are not provided to the LLMs as prompts.** The only prompts given to the models are three training input–output pairs and the test input grid, together with an official instruction [2]  directing models to perform inductive reasoning. The rules in Figure 8 are displayed solely to help readers better understand the underlying transformation. Therefore, the concerns you raised—such as “red dot vs. red square” or unspecified reflective obstacles—do not affect model evaluation, since these rules are never input to models.
>
> Nonetheless, we sincerely appreciate your correction. Indeed, for this kind of official abstract reasoning visualization[2,3], describing the element as a ‘red square’ is more suitable.  And we apologize that the rules in Figure 8 are overly brief. Below, we provide a detailed explanation of these rules.
>
> ### **(2) Clarifications on question design and ground-truth validity**
>
> **Second, we confirm that all tasks are designed to ensure a unique correct solution, and no task admits multiple correct solutions.** We apologize for any confusion, as we just put a concise explanation of the rules in original Figure 8 due to page limitations. Now we detailedly explain the complete latent rules in Figure 8:
>
> - The detailed latent rule of the level-3 task: Starting from the red square, plan a path that connects the blue squares. At each step, select the nearest blue square that is located directly above, below, to the left, or to the right of the current position; diagonal connections are not allowed.
> - The detailed latent rule of the level-4 task: Based on the given initial light, use physical principles to construct its multi-reflection trajectory. The light will reflect upon hitting the blue obstacle, whereas it will be absorbed when finally encountering the red obstacle.
>
> We hope that these detailed rule explanations could address your concerns. **And we have updated Figure 8 in the revised paper.** All of these latent rules were carefully designed to be diverse and challenging, with each rule admitting a unique solution.

---

> ### Author Response · Authors · 2025-11-21
> **Response Part 3**
>
> ### **W4&Q6: Validation to verify that increasing task complexity indeed reflects higher cognitive difficulty.**
>
> Thank you for your careful reading and opinion. Regarding the human accuracy across cognitive levels, we provide clarifications from three perspectives:
>
> **(1) Firstly, the statement in our paper is “human accuracy also generally decreases as the level increases”, is drawn from the result in Table 1**, where the human average accuracy of each level shows a modest downward tendency as the level increases. And the word “generally” indicates an overall average trend rather than a strict monotonic decrease.
>
> **(2) Secondly, our tasks were designed strictly following well-established cognitive theories in psychology[4]**, where the four levels and sub-rule types form a coherent hierarchical framework. Therefore, examining results by excluding a specific task (e.g., sorting) from its level is not appropriate to align with the established cognitive hierarchy. Here, regarding each level as a whole is more appropriate.
>
> **(3) Finally, as shown in the last two rows of Table 1, although human performance remains relatively stable, the decreasing trend is more noticeable and clearer on models.** We speculate that this may be because humans possess a higher ability in basic cognitive levels, which current models have not yet achieved. This observation is consistent with previous studies: [5] finds that even advanced models (e.g., GPT-4) only match novice-level human cognition, leaving room for improvement;  [6] finds that current models have not yet reached adult human-level performance in many 'cognitive' modules, such as commonsense reasoning, spatial cognition, and causal understanding. That’s potentially why human performance in our experiments remains relatively stable.
>
> ### **References**
>
> [1] Appelle S. (1972). Perception and discrimination as a function of stimulus orientation: the "oblique effect" in man and animals. *Psychological bulletin*, *78*(4), 266–278. https://doi.org/10.1037/h0033117.
>
> [2] Chollet, F. (2019). On the measure of intelligence. *arXiv preprint arXiv:1911.01547*.
>
> [3] Wu, J., Yu, M., Liu, L., Yeung, D. Y., & Zhou, J. (2025). Understanding LLMs' Fluid Intelligence Deficiency: An Analysis of the ARC Task. *arXiv preprint arXiv:2502.07190*
> [4] Primi, R. (2001). Complexity of geometric inductive reasoning tasks: Contribution to the understanding of fluid intelligence. *Intelligence*, *30*(1), 41-70.
>
> [5] Wang, X., Yuan, P., Feng, S., Li, Y., Pan, B., Wang, H., ... & Li, K. (2025, April). Coglm: Tracking cognitive development of large language models. In *Proceedings of the 2025 Conference of the Nations of the Americas Chapter of the Association for Computational Linguistics: Human Language Technologies (Volume 1: Long Papers)* (pp. 73-87).
>
> [6] Wang, X., Yuan, P., Feng, S., Pan, B., Li, Y., Sun, B., ... & Li, K. Tracking Cognitive Development of Large Language Models.

---

> ### Author Response · Authors · 2025-11-27
> **Sincerely looking forward to your reply**
>
> Dear Reviewer mbfn,
>
> We sincerely appreciate your time and effort in reviewing our work. We thank your deep comprehension and suggestions, which may have strengthened our manuscript.
>
> To address the concerns you raised, we provide **(1)** the criteria guided the selection of models in ablation study; **(2)** more detailed and refined experiment of Figure 5-*model accuracy versus variance*, Figure 6-*Impact of the Number of In-context Learning Samples*, and Figure 7- *Impact of the Inference Time*; **(3)** clarifications on ambiguous or mislabeled prompts; **(4)** explaination on question design and ground-truth validity, and we further updated the rule explaination of Figure 8; **(5)** illustration about the human accuracy across cognitive levels. Following your valuable comments, we have updated the content in the revised paper.
>
> We would appreciate the opportunity to continue dialogue with you to fully meet your expectations. If you still remain unclear about some aspects of our work or have other additional concerns, We are always prepared to discuss or provide more information to you. Our commitment is to address any concerns you may have and enhance the quality of our paper.
>
> Best regards.

---

### Official Review · Reviewer_cJQv · 2025-11-01

**Soundness:** 3
**Presentation:** 3
**Contribution:** 3
**Rating:** 6
**Confidence:** 3

**Summary:**

This work introduces DRE-Bench, a benchmark designed to evaluate dynamic reasoning through a hierarchical cognitive framework including 36 reasoning tasks. It frames fluid intelligence as the ability to generalize beyond memorized knowledge and reason, and designs rule-based code generators and solvers for each task. Empirical evaluations across different LLMs show that while models perform well on lower-level reasoning, they struggle with higher cognitive levels, indicating a gap from true fluid intelligence.

**Strengths:**

1. The task design is well aligned with cognitive psychology, clearly reflecting different types of reasoning and the varying levels of intelligence required for each task.

2. The proposed benchmark is a valuable contribution to the evaluation community, offering diverse and controllable challenging tasks.

3. The paper provides a thorough evaluation and analysis across multiple LLMs, shwoing how accuracy and stability vary with task complexity and cognitive level.

**Weaknesses:**

While the paper proposes a cognitively inspired hierarchy of reasoning tasks, it does not measure the time humans take to solve them. Given the small-scale human evaluation and potential variance, reporting the average solving time per task would better substantiate the claimed cognitive alignment and reveal the true difficulty gradient.

**Questions:**

In Table 1, why does o3-mini perform significantly better than other models on the Level-4-Mechanics tasks, achieving 31.75% accuracy while most other models show near-zero performance?

---

> ### Author Response · Authors · 2025-11-21
> **Response Part 1**
>
> Dear Reviewer cJQv,
>
> Thank you for appreciating our paper as a clear and thorough evaluation that offers diverse and controllable tasks. Your constructive comments and suggestions are valuable to us. Below is our detailed response to address your concerns.
>
> ### **W1: Reporting the average solving time per task.**
>
> Thanks for your valuable suggestion. As you suggested, reporting the average solving time of each task could improve the credibility of its cognitive alignment. Therefore,  we utilized the data records from previous human questionnaires to calculate the average time cost.
>
> **Specifically**, based on the dwell time on each question in the questionnaire, we computed the average solving time for all participants across each task. **Table A** presents the average solving time and accuracy of human participants for tasks on different levels. We can observe that, despite some fluctuations, the average human solving time generally increases as the task level becomes higher. This could further substantiate our claimed cognitive alignment and reveal the underlying difficulty gradient. However, we also note that for tasks with lower accuracy, the time spent by participants does not decrease too much. This indicates that participants engaged in continuous reasoning and attempts, rather than giving up quickly when the tasks were challenging, further substantiating our human study. **And we have added the table in Appendix E.9.3 of the revised paper.**
>
> Table A: The average time human participants take to solve each task, together with the average accuracy. (in seconds)
>
> |  | level-1 |  |  | level-2 |  |  | level-3 |  |  | level-4 |  |  |
> | --- | --- | --- | --- | --- | --- | --- | --- | --- | --- | --- | --- | --- |
> |  | Size | Count | Shape | Rotation | Move | Symmetry | Category | Sort | Planning | Optics | Mechanics | Thermal |
> | **Accuracy-avg** | 75.56 | 82.22 | 68.89 | 91.11 | 75.56 | 46.67 | 73.33 | 24.44 | 88.89 | 46.67 | 77.78 | 17.78 |
> | **Time-avg** | 73.4 | 63.4 | 62.2 | 90.2 | 88.8 | 93.0 | 109.8 | 106.4 | 105.8 | 165.8 | 132.1 | 147.3 |

---

> ### Author Response · Authors · 2025-11-21
> **Response Part 2**
>
> ### **Q1: Why does o3-mini perform significantly better than other models on the Level-4-Mechanics tasks?**
>
> Thank you very much for your careful reading. As you noticed, *o3-mini* indeed achieves the highest accuracy on the Level-4-Mechanics tasks in Table 1. Regarding this result, we conduct an in-depth survey and present additional experimental analyses.
>
> **(1) First, we conduct a comprehensive survey of prior research [1,2,3,4] on physical reasoning of large language models.**
>
> We find that the o3 series models have consistently demonstrated strong capability in physical understanding, as well as the mechanics sub-category in physics. The following studies provide supporting evidence on the strong physical reasoning capabilities of the o3 series.:
>
> - **<PHYBench: Holistic Evaluation of Physical Perception and Reasoning in Large Language Models>** [1]: The paper introduces PHYBench, 500 original physics problems ranging from high school to physics olympiad difficulty. It’s Figure 1 reports the accuracy and SEED score on PHYBench. It can be seen that o3-high, o3-mini-high, and o3-mini respectively rank 2nd, 5th, and 7th in overall accuracy among the 19 evaluated models, demonstrating the strong physical reasoning capabilities of the o3 series models. Moreover, even the highest-scoring LLMs still fall short of Human Experts in this paper, which is consistent with the conclusion of our paper.
> - **<Scaling Physical Reasoning with the PHYSICS Dataset>** [2]: The paper introduces PHYSICS, a dataset containing 16,568 high-quality physics problems spanning subjects and difficulty levels. It’s Table 7 shows the main results evaluated by accuracy. We observe that, compared to all other LLMs (Claude 3.7, Deepseek-R1, Qwen3-32B, etc), o3-high achieves the highest accuracy in almost all physical reasoning tasks, including various physical subjects,  different difficulties, and different languages. What’s more, on the specific Mechanics physical subject, o3-high also ranks the highest.
> - **<CMPhysBench: A Benchmark for Evaluating Large Language Models in Condensed Matter Physics>** [3]: The paper introduces CMPhysBench to assess the proficiency of LLMs in condensed matter physics. It’s Figure 5 presents the performance of LLMs on CMPhysBench. We can see that both o3-mini and o3 achieve high SEED scores and expert-labeled accuracy, ranking among the relatively top positions across all evaluated LLMs.
> - **<Humanity’s Last Exam>** [4]: HLE is a benchmark of 2,500 extremely challenging questions from dozens of subject areas, and its Table 3 reveals the category-wise breakdown of model performance. The upper half of its Table 3 shows the model scores under the Text-only setting on HLE, where o3-mini-high achieved the highest score of 15.3 in the Physics category, far surpassing all other reported models (Deepseek-R1, o1).
>
> **(2) Second, the other models are not entirely failing to understand the Level-4 tasks.**
>
> - **On one hand,** as the 0/1 accuracy in **Table 1**, other models like Claude-3.7-sonnet and o1 also perform reasonably well.
> - **On the other hand,** we have further calculated two finer-grained metrics (grid size precision and grid matching percentage) on model performance in the **Appendix Table 5**.
>
>     In detail, the grid size precision checks if the LLM's output grid size matches the ground truth grid [6]. If matching scores 1; otherwise, it scores 0. This assesses the model's ability to handle grid dimensions. And the grid matching percentage means the proportion of matching elements between the response grid and ground truth grids. This percentage offers a finer-grained score. According to the Level-4-Mechanics task in **Appendix Table 5**, we find that although other models attain low 0/1 accuracy, their scores on grid size precision and grid matching percentage are not substantially worse than those of o3-mini.  For example, Claude-3.7, DeepSeek-R1, and O1 all achieve the same grid-size precision (100) as o3-mini, and their grid-matching percentages are comparable to that of o3-mini. Claude-3.7 (75) even attains a higher grid-matching percentage than o3-mini (73). This suggests that other models can partially infer and apply the physical mechanics rule,  and that they do possess a non-negligible degree of understanding and reasoning ability for this task.

---

> ### Author Response · Authors · 2025-11-21
> **Response Part 3**
>
> **(3) In summary, we think the good performance of o3-mini on the Level-4-Mechanics tasks is a relatively rational phenomenon for the following points:**
>
> - Prior evaluation works[1,2,3,4] have consistently shown that o3-mini excels at physical and mechanics-related reasoning, which aligns with our findings;
> - Our finer-grained metrics in the **Appendix Table 5** indicate that other models do demonstrate partial understanding of the Level-4-Mechanics tasks;
> - Our experiments strictly follow the official abstract reasoning evaluation settings[7], where each model was evaluated three independent times, ensuring the reliability and confidence.
>
> ### **References**
>
> [1]  Qiu, S., Guo, S., Song, Z. Y., Sun, Y., Cai, Z., Wei, J., ... & Zhu, H. X. (2025). Phybench: Holistic evaluation of physical perception and reasoning in large language models. *arXiv preprint arXiv:2504.16074*.
>
> [2] Zheng, S., Cheng, Q., Yao, J., Wu, M., He, H., Ding, N., ... & Ye, P. (2025). Scaling physical reasoning with the physics dataset. *arXiv preprint arXiv:2506.00022*.
>
> [3] Wang, W., Huang, D., Li, J., Yang, T., Zheng, Z., Zhang, D., ... & Weng, H. (2025). CMPhysBench: A Benchmark for Evaluating Large Language Models in Condensed Matter Physics. *arXiv preprint arXiv:2508.18124*.
> [4] Phan, L., Gatti, A., Han, Z., Li, N., Hu, J., Zhang, H., ... & Wykowski, J. (2025). Humanity's last exam. *arXiv preprint arXiv:2501.14249*.
>
> [5] Madaan, A., Tandon, N., Gupta, P., Hallinan, S., Gao, L., Wiegreffe, S., ... & Clark, P. (2023). Self-refine: Iterative refinement with self-feedback, 2023. *URL https://arxiv. org/abs/2303.17651*.
>
> [6] Wu, J., Yu, M., Liu, L., Yeung, D. Y., & Zhou, J. (2025). Understanding LLMs' Fluid Intelligence Deficiency: An Analysis of the ARC Task. *arXiv preprint arXiv:2502.07190*.
>
> [7] Chollet, F. (2019). On the measure of intelligence. *arXiv preprint arXiv:1911.01547*.

---

> > ### Author Response · Authors · 2025-11-27
> > **Sincerely looking forward to your reply**
> >
> > Dear Reviewer cJQv,
> >
> > Thank you for your review time and insightful comments on our paper. We have provided corresponding responses and results in great detail. If you have any other concerns, we are more than happy to provide additional clarification as well as experiments at any time. Sincerely looking forward to your reply!
> >
> > Best,
> >
> > Authors

---

### Author Response · Authors · 2025-11-21
**General Response： Thanks, Contributions, New Clarifications, and New Experiments**

Dear Program Chairs, Senior Area Chairs, Area Chairs, and Reviewers:

Thanks for your attention to this message. We sincerely thank all reviewers for their time and effort in reviewing our paper. We are encouraged that the reviewers generally recognize the value of our proposed DRE-Bench. We hope that DRE-Bench will enable fine-grained and meaningful evaluations of models’ reasoning abilities corresponding to specific levels of human-like intelligence.

### **Contributions**

The main contributions recognized by reviewers are concluded as followed:

- **A Cognition-aware Abstract Reasoning Benchmark [cJQv, mbfn, QK7j, K9CV]:** We propose an abstract reasoning benchmark with a cognition hierarchy, providing a more structural and comprehensive system to analyze the LLMs' true fluid intelligence.
- **A verifiable and scalable data generation engine [mbfn, QK7j, K9CV]:** We develop a verifiable and scalable data engine to dynamically generate abstract reasoning data with various complexities, by designing a generator and solver for each task.
- **Comprehensive Evaluations and interesting findings[cJQv, mbfn]:** We perform comprehensive evaluations on a variety of popular LLMs, indicating that the existing LLMs still struggle to solve the reasoning problem of high cognitive levels, highlighting that they remain far from achieving true fluid intelligence.

### **New Clarifications, and New Experiments**

We also thank all reviewers for their insightful and constructive suggestions, which helped a lot in further improving our paper. In addition to the pointwise responses below, we summarize supporting clarifications and experiments added in the rebuttal according to the reviewers’ suggestions.

**(1) New Clarifications**

- Appendix E.2 Table 5 contains two more fine-grained evaluation metrics; [K9CV, cJQv]
- The criteria that guide the selection of models in the ablation study; [mbfn]
- Clarification on question design and the unique ground-truth; [mbfn]

**(2) New Experiments**

- Refined experiments to further validate the ablation study, in Figure 5, 6, 7; [mbfn]
- Detailed design of the human Experiment, including the age distribution, significance t-testing, salary of participants, and ui interface of the questionnaire;  [QK7j]
- The average time human participants take to solve each task; [cJQv]
- Experiment of combining visual input with CoT cues; [K9CV]
- Detailed Analysis of Model Failure Modes; [QK7j]

We hope our pointwise responses below can clarify all reviewers’ confusion and alleviate all concerns. Thanks to all reviewers, and **we have incorporated some of the mentioned contents into the revised paper, highlighted in blue**.

---

### Author Response · Authors · 2025-12-02
**Letter to Area Chair—Part 1**

Dear Area Chair,

Considering this unforeseen incident, we fully recognize the challenges confronting the ICLR organizing committee. To facilitate your workflow, we have summarized this letter about our work, including the main contributions, key rebuttal points, and score changes. A detailed statement is as follows:

## **Summary of Our Work**

This paper introduces a Dynamic Reasoning Evaluation benchmark, DRE-Bench, designed to assess the genuine fluid intelligence of large language models by the abstracting reasoning task. The principal contributions and insights are summarized as follows:

1. **Cognitive hierarchy.** We propose an abstract reasoning benchmark with a cognition hierarchy, providing a more structured and comprehensive system to analyze the LLMs' true fluid intelligence.
2. **Scalable data engine.** We develop a verifiable and scalable data engine to dynamically generate abstract reasoning data with various complexities, by designing a generator and solver for each task.
3. **General performance.** Existing LLMs still struggle to solve the reasoning problem of high cognitive levels. As the cognitive level of the reasoning tasks increases, model accuracy consistently declines (particularly for tasks involving physical concepts), and the difference between models also becomes more pronounced.
4. **Impact factors.** In abstract reasoning tasks, increasing the number of in-context training examples can slightly boost LLMs' performance. While inference time scaling plays a more important role in low-level reasoning tasks, but may be insufficient towards high-level latent rules as complexity increases. However, adding visual information has little positive impact on model accuracy, and sometimes even leads to a minor decrease.
5. **Interesting findings.** Compared to humans, current LLMs may demonstrate a distinct understanding of spatial orientation. LLMs achieve higher and more consistent accuracy in vertical (up/down) directions than in horizontal (left/right) directions in moving tasks. But in symmetry tasks, LLMs perform better horizontally than vertically.

## **Response to Reviewers' Comments**

### **Response to Reviewer cJQv**

| Questions | Response | Addressed or Not? |
| --- | --- | --- |
| Reporting the average solving time per task would better substantiate cognitive alignment. | We provided a comprehensive experiment on the average time human participants take to solve each task.  And incorporated this result **into the revised paper Appendix E.9.3.** | non-responsive |
| Why does o3-mini perform significantly better than other models on the Level-4-Mechanics tasks? | We conduct an in-depth survey and experiments to explain this phenomenon. We provide conclusions of prior works, finer-grained metrics in the Appendix, and experiment settings to further illustrate. | non-responsive |
|  |

### **Response to Reviewer mbfn**

| Questions | Response | Addressed or Not? |
| --- | --- | --- |
| What criteria guided the selection of models in Figures 5–7 and Table 2? | We detailedly illustrate the reasons for model selection in Figure 5–7 and Table 2. We further provided more comprehensive experiments in appendix and updated the **Figure 6, 7 in the revised paper**. | non-responsive |
| How were ground truths defined for multi-path problems, and were alternative valid solutions considered during evaluation? | We clarify that all of our task designs guarantee a unique correct solution.  | non-responsive |
| How were ambiguous or mislabeled prompts (e.g., “red dot” vs. “red square,” unspecified reflective obstacles) handled in scoring and analysis? | We clarify that the “rules” shown in Figure 8 are not provided to the LLMs as prompts, and are not related to the experimental results. We also write more detailed rules in **Figure 8**. |  |
| How to justify the statement that accuracy “generally decreases” with increasing cognitive level? | We provided a detailed validation of the statement from three perspectives: average accuracy, well-established cognitive hierarchy, and analysis aligned with previous works.  | non-responsive |
|  |

---

> ### Author Response · Authors · 2025-12-02
> **Letter to Area Chair—Part 2**
>
> ### **Response to Reviewer QK7j**
>
> | Questions | Response | Addressed or Not? |
> | --- | --- | --- |
> | Why were only 20 annotators selected as the sample for the human comparison experiment?  | We enlarged the scale of human participants to 40, and update all the statistical data and results of human study. | Reviewer replied: issue addressed, and believe this is a novel piece of work. |
> | There is a lack of detailed age distribution and significance testing. | We provide detailed age distribution of participants, and the t-testing between human and model performance, validating our conclusion. As **Figure 17, Table 8, Table 9 of the revised paper.** | Reviewer replied: issue addressed, and believe this is a novel piece of work. |
> | How was the seriousness of participants and the validity of the questionnaire ensured?  | We have provided a salary of 30 $ per hour to each participant and salary and given them detailed instructions. | Reviewer replied: issue addressed, and believe this is a novel piece of work. |
> | Were potential cognitive biases or other influencing factors considered during the questionnaire design process?  | We carefully designed the user-friendly UI interface for the questionnaire, and have open-sourced the UI code at https://anonymous.4open.science/r/DRE-Bench-8098.  | Reviewer replied: issue addressed, and believe this is a novel piece of work. |
> | What are the specific failure modes of the models in high-level cognitive tasks? Are there certain types of errors that repeatedly occur? | We provide a more detailed analysis of model failure modes in **Figure 16 of the revised paper.** We compare errors made by different models on the same task and summarize some common error patterns, especially in high levels. | Reviewer replied: issue addressed, and believe this is a novel piece of work. |
> | Key Findings should be deeper. | We emphasized that beyond the findings in introduction, extra interesting observations are revealed in the experiments section. And other reviewer recognized the novelty of our contributions and findings. | Reviewer replied: issue addressed, and believe this is a novel piece of work. |
> | Insufficient Analysis of Visual Information Impact. | We clarify that visual information is not necessary in this task. And our two visualization formats are carefully designed according to the official format. | Reviewer replied: issue addressed, and believe this is a novel piece of work. |
> |  |
>
> ### **Response to Reviewer K9CV**
>
> | Questions | Response | Addressed or Not? |
> | --- | --- | --- |
> | This paper only uses the all-right/all-wrong grid matching criterion, limiting the fine-grained analysis of model capabilities. | We explain that we have provided more fine-grained metrics in **the appendix E.2** and show more detailed analysis. | Reviewer replied: issue addressed, and consider this work as solid and insightful. |
> | What happens if combining visual input with CoT cues? | We conduct new experiments to further investigate the impact of visual input with CoT. And we have updated the results of **Table 2 in the revised paper.** | Reviewer replied: issue addressed, and consider this work as solid and insightful. |
> |  |
>
> ## **Score Changes**
>
> The original scores of our paper are  6, 6, 2, 2. **Before the cut-off incident, the scores had been increased to 6, 6, 4, 2.** Two reviewers [QK7j,  K9CV] responded to our rebuttal and acknowledged that their concerns were fully or generally addressed. Among them, [QK7j] increased the scores **from 2 to 4**, while [K9CV] maintained a positive score of 6 and **increased the confidence**. The remaining two reviewers [cJQv] and [mbfn] have not yet responded. Among them, reviewers [cJQv] initially gave a positive score of 6. Nevertheless, we have provided a point-by-point response to the comments and believe this adequately addresses the reviewers' concerns. **And we have revised the paper according to the reviewers' comments.**
>
>
> ## **Final Conclusion**
>
> Even though this unexpected incident occurred, we can affirm that we have strictly adhered to all double-blind review regulations, and we maintain a meticulous attitude all the time in addressing the reviewers’ concerns. We fully recognize the challenges faced by the ICLR organizing committee, and we hope this summary may help to alleviate your workload. We respectfully request that you consider accepting our work, and we will continue contributing to fairness and progress in the academic community.

---

### Meta-Review · Area_Chair_KrQx · 2026-01-06

**Summary:**

The paper proposes DRE-Bench, a benchmark designed to evaluate fluid intelligence in LLMs. While reviewers acknowledged the value of the scalable data engine and the study's interesting findings, significant concerns were raised during the initial review. The authors made commendable efforts to address these issues during the rebuttal; however, the resulting updates represent fundamental changes to the paper's core validation that are too extensive to be adequately verified within the constraints of the current review cycle.    The authors are encouraged to submit the refined manuscript to another venue.

**Reviewer Concerns:**

Remaining concerns from negative reviewers (projected by the AC):

1. Experiment design with human evaluation

2. Further validation is needed to ensure the generated items reflect increasing cognitive difficulty rather than artifacts of task design.

**Reviewer Scores:**

6/6/4/4

---

### Decision · Program_Chairs · 2026-01-26

Reject